# Separability Analysis for Causal Discovery in Mixture of DAGs

**Burak Varıcı**                                                                *varicb@rpi.edu*
*Electrical, Computer, and Systems Engineering*
*Rensselaer Polytechnic Institute*

**Dmitriy Katz-Rogozhnikov**                                         *dkatzrog@us.ibm.com*
*IBM Research AI*

**Dennis Wei**                                                              *dwei@us.ibm.com*
*IBM Research AI*

**Prasanna Sattigeri**                                                  *psattig@us.ibm.com*
*IBM Research AI*

**Ali Tajer**                                                            *tajer@ecse.rpi.edu*
*Electrical, Computer, and Systems Engineering*
*Rensselaer Polytechnic Institute*

**Reviewed on OpenReview:** *https://openreview.net/forum?id=ALRWXT1RLZ*

## Abstract

Directed acyclic graphs (DAGs) are effective for compactly representing causal systems and specifying the causal relationships among the system's constituents. Specifying such causal relationships in some systems requires a mixture of multiple DAGs – a single DAG is insufficient. Some examples include time-varying causal systems or aggregated subgroups of a population. Recovering the causal structure of the systems represented by single DAGs is investigated extensively, but it remains mainly open for the systems represented by a mixture of DAGs. A major difference between single- versus mixture-DAG recovery is the existence of node pairs that are separable in the individual DAGs but become inseparable in their mixture. This paper provides the theoretical foundations for analyzing such inseparable node pairs. Specifically, the notion of *emergent edges* is introduced to represent such inseparable pairs that do not exist in the single DAGs but emerge in their mixtures. Necessary conditions for identifying the emergent edges are established. Operationally, these conditions serve as sufficient conditions for separating a pair of nodes in the mixture of DAGs. These results are further extended, and matching necessary and sufficient conditions for identifying the emergent edges in tree-structured DAGs are established. Finally, a novel graphical representation is formalized to specify these conditions, and an algorithm is provided for inferring the learnable causal relations.

## 1 Introduction

### 1.1 Motivation

Causal Bayesian networks can effectively represent causal relationships among random variables generated in a network of interacting agents. Such a network can be specified by, for instance, a directed acyclic graph (DAG) and a joint probability distribution that factorizes with respect to the DAG. Causal discovery is the process of using observational data and inferring a DAG's causal structure, i.e., its edges' locations and orientations. One common approach to causal discovery involves performing statistical tests for determining conditional independence relationships in the observation data. When the system can be represented by a DAG, the inferred independence relationships are subsequently converted

into DAG ($d$-separation) separation statements to infer the structure (Spirtes et al., 2000; Colombo et al., 2012). These approaches are known as constraint-based methods. Other notable approaches include score-based methods (Chickering, 2002; Tsamardinos et al., 2006), hybrid methods (Nandy et al., 2018), and continuous-optimization-based methods (Zheng et al., 2018).

A widely used assumption in causal discovery is that the observed data is independent and identically distributed (i.i.d.) according to one DAG. This is not valid in a wide range of settings and applications in which more complex mechanisms generate the observed data. Such complex data can be modeled more accurately by multiple distinct causal models on the same set of variables. For instance, in fMRI data analysis, different brain regions interact differently under various activities of the subjects (Neumann et al., 2010). Another example is gene expression data from certain diseases, like ovarian cancer, which comprises multiple subtypes that are challenging to distinguish (Reid et al., 2017). In a successful gene-editing technique CRISPR-Cas9 (Ran et al., 2013), Cas9 enzyme cuts the DNA at a specific location. However, depending on an array of factors, such as the delivery method of the gene-editing tool and the complexity of the genetic alteration needed, the gene-editing tool is not always successful, resulting in ambiguity while grouping gene expression samples based on the corresponding distribution. In psychological studies, often multiple time-series trajectories (e.g., depression-related symptoms over a period of time) are collected from different patients (Bulteel et al., 2016). This data is fitted better with multi-modal linear dynamic systems instead of a single model, and labels of the trajectories – model membership of the samples – are unknown to the researcher. For more examples of mixture models in dynamical systems, we refer to Chen & Poor (2022). Furthermore, in some settings, the causal models are cyclic. For instance, the processes that regulate hormones without reaching an equilibrium state (Pirahanchi et al., 2023). The statistical models of the data in such heterogeneous systems cannot be represented with a single DAG and require a mixture of them.

## 1.2   Contribution

Despite the significance of causal discovery from complex models, it is far less investigated compared to causal discovery in single DAGs. The advances in the discovery of single DAGs are due to well-specified frameworks that facilitate formalizing various causal discovery objectives and are amenable to tractable analysis. However, the counterparts of such frameworks for more complex models are not well-established in the literature. This paper formalizes a framework that facilitates addressing causal discovery objectives in systems represented by a mixture of DAGs. The first important step towards this goal is establishing *possibility* results, that is, identifying and discerning the causal relationships that can possibly be learned from those that are impossible to learn. It can be readily verified that some natural choices generally investigated in single DAGs are impossible when working with mixture models. For instance, mixing the models renders learning the constituent DAGs' structures impossible. Similarly, recovering the union of the edges of individual DAGs is ill-posed without additional information or assumptions on the causal models. To address such learnable versus unlearnable dichotomy to develop the key ideas, we consider a mixture of two DAGs.

We formalize the separability framework in which the inferential decisions are centered around *separability analysis*, which refers to identifying the pair of random variables that cannot be made conditionally independent in the mixture distribution. Accordingly, we refer to the node pairs associated with inseparable random variables as *inseparable* pairs. We introduce proper edge notations to visualize such edges graphically. Given this framework, we investigate what can be learned about the underlying DAGs in a mixture model. We adopt a constraint-based approach and use conditional independence (CI) tests on the mixture distribution to identify inseparable node pairs. We note that *faithfulness* – conditional independencies in the data being due to the separations in the true DAG – is a core assumption in constraint-based causal discovery (Spirtes et al., 2000; Pearl, 2009). Hence, we adopt a *mixture faithfulness* assumption in this paper, similar to the existing literature on causal discovery in a mixture of DAGs. Once the inseparable node pairs are identified, we construct a *mixture graph* with these node pairs as its edges. The main challenge in interpreting the mixture graph is the existence of node pairs that are not adjacent (connected) in all individual DAGs but are not conditionally independent in the mixture distribution for any conditioning set. Due to their importance, we analyze this subset of inseparable pairs and refer to them as *emergent edges*. We observe that the set of nodes with varying conditional distributions across the individual DAGs, denoted by $\Delta$, plays a critical role in forming inseparable pairs in the mixture. We start by investigating a mixture of two arbitrary DAGs and establish conditions under which any arbitrary pair of nodes in the mixture model is separable. Equivalently, for a node pair to form an emergent edge, these

conditions necessarily do not hold. Furthermore, we specialize these results to *tree-structured*[1] DAGs, for which we establish the necessary and sufficient conditions for node pairs to be separable. Our main contributions are as follows.

- We introduce a *mixture graph* to represent the inseparable pairs of nodes in the mixture distribution. To interpret the edges of the mixture graph, we present a sufficient condition for two nodes to be separable. This result sheds light on the inadequacy of an existing graphical representation in the literature.
- We strengthen the results for a mixture of tree-structured DAGs and establish necessary and sufficient conditions for two nodes to form an *emergent edge* in the mixture.
- We introduce new edge notations for inseparable node pairs to represent the above conditions and partially orient the skeleton of the mixture graph. We discuss the inference of these oriented edges from unshielded triples and show that v-structures upon nodes in the $\Delta$ set can be recovered. Finally, we devise an algorithm built on our theoretical results to recover the learnable causal relationships.

## 1.3 Related Work

**Causal discovery of a mixture of DAGs.** There are three studies closely related to this paper. The shared objective of these papers is to develop a graphical model to represent as many CI relationships in the mixture distribution as possible. Spirtes (1995) proposes a fused graph and shows that the mixture distribution is with respect to the fused graph. However, it does not provide separability theorems for a pair of nodes. Strobl (2023) proposes a mixture graph such that the mixture distribution is Markov with respect to the proposed graph. However, this study does not provide counterparts of our separability results either. It proposes an algorithm that is designed exclusively for time-series data. In the work most closely related to this paper, the study by Saeed et al. (2020) considers a mixture of DAGs and proposes a composite DAG that satisfies the Markov property. Subsequently, it represents the mixture distribution with a maximal ancestral graph (MAG) to use the existing structure learning algorithms directly. This approach is only valid under a strict assumption called *poset compatibility* that rules out any conflicting ancestral relationships across individual DAGs (e.g. cycles). Importantly, none of these studies provide conditions for when an emergent edge arises. In this paper, we mainly focus on tree-structured component DAGs, but we do not restrict the topological order of the individual DAGs and do not consider time-series data.

**Causal discovery from multiple clusters/contexts.** Another approach to our problem is to divide it into two independent stages: clustering the samples and then performing structure learning on each cluster individually (Zhang & Glymour, 2020; Chen et al., 2021). Learning from multiple contexts is a well-studied topic in the causality literature, especially in interventional (experimental) settings (Huang et al., 2020; Zhang et al., 2017; Mooij et al., 2020; Jaber et al., 2020). In these studies, the context of an observed sample, i.e., which graph it belongs to, is assumed to be known in contrast to observing a mixture distribution. In another line of work, several algorithms perform causal discovery of cyclic causal models by assuming that the observed distribution is the equilibrium state of multiple involved causal models (Forré & Mooij, 2017; Bongers et al., 2021; Améndola et al., 2020; Ghassami et al., 2020). The study in Winn (2012) addresses the problem of representing context-specific independence via leveraging factor graphs with interventions. This is an interesting yet underexplored approach compared to the graphical modeling of interventions based on (Pearl, 2009). The study in Thiesson et al. (1998) aims to learn the parameters of multiple Bayesian networks from combined data. Specifically, heuristic score-based algorithms are proposed for jointly learning the structure and parameters of multiple Bayesian networks. Finally, Meila & Jordan (2000) consider learning a discrete mixture-of-trees distribution. However, the considered model is not causal.

**Applications of tree-structured DAGs.** Causal trees are shown to be computationally effective while still being useful to closely approximate more complex models (Acid & de Campos, 1994). Beyond their convenience for providing tractable analysis, they are widely studied in causal discovery literature and have real applications, e.g., in biological networks. Specifically, protein signaling pathways are commonly modeled by causal trees. For instance, bi-partite causal graphs are used to model gene networks, in which genes induce protein expressions, and the expressed proteins either inhibit or activate other genes (Kontou et al., 2016). NF-kB protein signaling pathway, which activates mammalian immune system cells to produce antibodies against inflammation, is also modeled by causal trees (Lodish, 2016).

---

[1]Specifically, a tree-structured DAG is a DAG whose underlying undirected graph is a tree. This structure can be referred to as *directed tree*, *polytree* or *singly connected network* as well.

## 2 Separability Framework for DAG Mixtures

**DAG Models.** In this section, we provide a DAG mixture model and introduce the notations needed for analyzing them. For clarity purposes, we initially focus on a mixture of two DAGs and discuss the generalization to a mixture of an arbitrary number of DAGs in Appendix B.

Consider two *component* DAGs $\mathcal{G}_1 \triangleq (\mathbf{V}, \mathbf{E}_1)$ and $\mathcal{G}_2 \triangleq (\mathbf{V}, \mathbf{E}_2)$ defined over the same set of nodes $\mathbf{V} \triangleq \{1, \dots, n\}$. $\mathbf{E}_\ell$ denotes the set of *directed* edges in graph $\mathcal{G}_\ell$, for $\ell \in \{1, 2\}$. We define $\mathrm{pa}_\ell(i), \mathrm{ch}_\ell(i), \mathrm{an}_\ell(i), \mathrm{de}_\ell(i)$, and $\mathrm{sp}_\ell(i)$ to denote parents, children, ancestors, descendants, and spouses (i.e., nodes that share a common child) of node $i$ in graph $\mathcal{G}_\ell$ for $\ell \in \{1, 2\}$, respectively. We denote the maximum in-degree of a node in either component DAG by $d$. We augment the set $\mathrm{pa}_\ell(i)$ by adding the node $i$ to it and denote the augmented set by $\mathrm{pa}_\ell^+(i) \triangleq \mathrm{pa}_\ell(i) \cup \{i\}$. Similarly, we also define the augmented sets $\mathrm{ch}_\ell^+(i), \mathrm{de}_\ell^+(i)$, and $\mathrm{an}_\ell^+(i)$. We extend these notations to any desired subset of nodes. Specifically, for any $A \subseteq \mathbf{V}$ we define

$$\mathrm{pa}_\ell(A) \triangleq \bigcup_{i \in A} \mathrm{pa}_\ell(i) \,, \tag{1}$$

and define $\mathrm{ch}_\ell(A), \mathrm{an}_\ell(A), \mathrm{de}_\ell(A)$, and $\mathrm{sp}_\ell(A)$ similarly.

**Probability Models.** The random variable associated with node $i \in \mathbf{V}$ is denoted by $X_i$, which can be continuous or discrete. Accordingly, we define the vector of variables $X \triangleq (X_1, \dots, X_n)^\top$. For any subset of nodes $A \subseteq X$, we define $X_A \triangleq \{X_i \: : \: i \in A\}$. We denote the probability distributions of $X$ under $\mathcal{G}_1$ and $\mathcal{G}_2$ by $p_1$ and $p_2$, respectively. These distributions factorize with respect to their associated DAGs according to

$$p_\ell(x) = \prod_{i \in [n]} p_\ell(x_i \mid x_{\mathrm{pa}_\ell(i)}) \,, \quad \forall \ell \in \{1, 2\} \,. \tag{2}$$

This factorization implies that $p_1$ and $p_2$ can be distinct even when $\mathbf{E}_1$ and $\mathbf{E}_2$ are identical. Hence, the differences between the two causal models should be defined through the nodes with varying causal mechanisms, i.e., conditional distributions. We denote the set of nodes that have distinct conditional distributions across two models by $\Delta$, i.e.,

$$\Delta \triangleq \{i \in \mathbf{V} : p_1(X_i \mid X_{\mathrm{pa}_1(i)}) \neq p_2(X_i \mid X_{\mathrm{pa}_2(i)})\} \,. \tag{3}$$

**Observations Mixture Model.** The observed data is generated by a mixture of distributions $p_1$ and $p_2$. The model generating the observation at a given instance is unknown. To formalize this, we define $L \in \{1, 2\}$ as a latent random variable that accounts for the true model underlying the observed data, where $L = \ell$ specifies that the true model is $p_\ell$. We denote the probability mass function (pmf) of $L$ by $q$. We assume that $L$ and $X$ are statistically independent. These render the following mixture distribution for the observed samples $X$.

$$p_\mathrm{M}(x) \triangleq \sum_{\ell \in \{1, 2\}} q(\ell) \cdot p_\ell(x) \,. \tag{4}$$

Next, we introduce the notations and definitions for formalizing the separability framework, formulating the causal discovery objectives, and describing the analytical results. Connectivities between node pairs, represented by edges, in a graphical representation of causal models are collectively denoted by the skeleton of the graph. An edge between two nodes essentially means that the random variables associated with the nodes cannot be made conditionally independent. Such nodes are equivalently (under faithfulness) referred to as *inseparable* nodes. The exact meaning and orientation of the edges can change across different representations, e.g., directed edges in DAGs and bidirected edges in MAGs, but the inseparability requirement remains the same.

The existing literature on studying a mixture of DAGs focuses on establishing a graphical representation that can (partially) represent conditional independence relations in the mixture distribution $p_\mathrm{M}$ (Spirtes, 1995; Saeed et al., 2020; Strobl, 2023). While effective for inferring the separability statements, the existing approaches do not represent some of the important aspects of the mixture models such as interpreting the cycles across the individual DAGs. We specify four directed and undirected graphical models, each serving a distinct purpose in the analysis. In these graphs, we use the following edge notations:

- $(i - j)$: the undirected edge from node $i$ to node $j$.
- $i \to j$: the directed edge from node $i$ to node $j$.
- $i \overset{\ell}{\to} j$: the directed edge from node $i$ to node $j$ in $\mathbf{E}_\ell$.
- $i \overset{\ell}{\rightsquigarrow} j$: denotes $i \in \mathrm{an}_\ell(j)$.

We call the tuple of nodes $(i, k, j)$ an *unshielded triple* in $\mathcal{G}_\ell$ if $(i - k)$ and $(j - k)$ are edges of $\mathcal{G}_\ell$ but $i$ and $j$ are not adjacent. If we further have $i \overset{\ell}{\to} k \overset{\ell}{\leftarrow} j$, node $k$ is called a *unshielded collider* in $\mathcal{G}_\ell$ and $(i \overset{\ell}{\to} k \overset{\ell}{\leftarrow} j)$ is called a *v-structure*. We introduce an auxiliary node denoted by $y$, representing the latent variable $L$ of the mixture model.

**Definition 1 (Union Graph)** *The union graph, denoted by $\mathcal{G}_U \triangleq (\mathbf{V}, \mathbf{E}_U)$, is an undirected graph constructed by setting $\mathbf{E}_U$ as the union of all edges in $\mathcal{G}_1$ and $\mathcal{G}_2$ after removing their orientations, i.e.,*

$$\mathbf{E}_U \triangleq \{(i - j) : i, j \in \mathbf{V}, \ \exists \ell \in \{1, 2\} : i \overset{\ell}{\to} j\} . \tag{5}$$

By recalling the definition of $\Delta$ in (3), next we define fused graphs.

**Definition 2 (Fused Graph (Spirtes, 1995))** *The fused graph $\mathcal{G}_F \triangleq (\mathbf{V}_F, \mathbf{E}_F)$ is a directed graph constructed by setting $\mathbf{V}_F \triangleq \mathbf{V} \cup \{y\}$, superposing the edge sets $\mathbf{E}_1$ and $\mathbf{E}_2$, and adding edges from $y$ to the nodes in $\Delta$, i.e.,*

$$\mathbf{E}_F \triangleq \{i \to j : i, j \in \mathbf{V}, \ \exists \ell \in \{1, 2\} : i \overset{\ell}{\to} j\} \cup \{y \to i : i \in \Delta\} . \tag{6}$$

A fused graph can contain cycles but does not necessarily encode the complete set of conditional independencies in mixture distribution. Next, we specify a composite DAG[2], defined by Saeed et al. (2020). The composite DAG involves concatenating $\mathcal{G}_1$ and $\mathcal{G}_2$ with a single-node graph $\mathcal{G}_y \triangleq (\{y\}, \emptyset)$ between them. Figure 1 shows an example of constructing a composite DAG from component DAGs. The composite DAG is formalized next.

**Definition 3 (Composite DAG (Saeed et al., 2020, Definition 3.1))** *Given graphs $\mathcal{G}_1$, $\mathcal{G}_2$, and $\mathcal{G}_y \triangleq (\{y\}, \emptyset)$, the composite DAG $\mathcal{G}_C$ is constructed by a series composition of $\mathcal{G}_1$, $\mathcal{G}_y$, and $\mathcal{G}_2$. The composite DAG $\mathcal{G}_C$ inherits the directed edges of $\mathcal{G}_1$ and $\mathcal{G}_2$ and additionally includes directed edges from node $y$ to nodes $\Delta$ in $\mathcal{G}_1$ and $\mathcal{G}_2$.*

The composite DAG consists of $2n + 1$ nodes. We refer to the node corresponding to node $i$ in $\mathcal{G}_\ell$ by $i_\ell$. Accordingly, for any $A \subseteq \mathbf{V}$ we define $\bar{A} \triangleq \{i_\ell : i \in A, \ \ell \in \{1, 2\}\}$.

**Definition 4 (Composite DAG $d$-separation)** *For any disjoint sets $A, B, C \subseteq \mathbf{V}$, we say that given $C$, $A$ and $B$ are $d$-separated in the mixture model when given $\bar{C}$, $\bar{A}$ and $\bar{B}$ are $d$-separated in $\mathcal{G}_C$. We denote such $d$-separation by $A \perp\!\!\!\perp^M B \mid C$.*

**Definition 5 (Mixture Global Markov Property)** *We say that mixture distribution $p_M$ satisfies the global Markov property with respect to $\mathcal{G}_C$ if for any disjoint sets $A, B, C \subseteq \mathbf{V}$ such that $A \perp\!\!\!\perp^M B \mid C$, we have $X_A$ and $X_B$ are conditionally independent given $X_C$. This is denoted by $X_A \perp\!\!\!\perp X_B \mid X_C$.*

**Definition 6 (Mixture Faithfulness)** *The converse of the Markov property between $\mathcal{G}_C$ and $p_M$ is called* mixture faithfulness*, that is $p_M$ is faithful to $\mathcal{G}_C$ if $X_A \perp\!\!\!\perp X_B \mid X_C$ in $p_M$ implies $A \perp\!\!\!\perp^M B \mid C$.*

Mixture faithfulness facilitates forming inferences on the graph structure from CI tests on distribution, and it is adopted by constraint-based causal discovery literature. In this paper, we work under the mixture faithfulness assumption. While the composite DAG is not the representation we are characterizing in this paper, it captures important conditional independence relations among $X_i$ and $X_j$, which facilitate our separability analysis. Next, we define *mixture graphs*, distinct from the composite DAGs defined above. The purpose of the mixture graph is to represent the inseparable node pairs compactly over nodes $\mathbf{V}$. In Section 4 we will analyze the properties of mixture graphs.

**Definition 7 (Mixture Graph)** *The mixture graph, denoted by $\mathcal{G}_M \triangleq (\mathbf{V}, \mathbf{E}_M)$, is a graph over $\mathbf{V}$ with undirected edges between the pair of nodes that are always dependent in the mixture distribution $p_M$, i.e.,*

$$\mathbf{E}_M \triangleq \{(i - j) : i, j \in \mathbf{V}, \ \nexists A \subseteq \mathbf{V} \setminus \{i, j\} : \ X_i \perp\!\!\!\perp X_j \mid X_A\} . \tag{7}$$

---

[2]Referred to as *mixture DAG* in (Saeed et al., 2020).

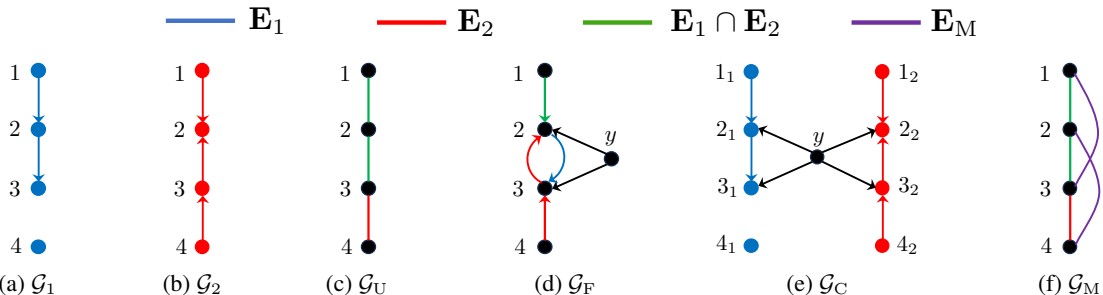

Figure 1: (a) and (b) are the component DAGs; (c) is the union graph; (d) is the fused graph; (e) is the composite DAG; and (f) is the mixture graph. Note that mixture graph $\mathcal{G}_M$ has emergent edges $\mathbf{E}_E = \{(1-3), (2-4)\}$. This example assumes that if parents of a node are invariant, then its conditional distribution is also invariant, and set $\Delta = \{2, 3\}$.

Clearly, if the pair $(i, j)$ is connected in at least one of the graphs $\mathcal{G}_1$ and $\mathcal{G}_2$, then $\mathbf{E}_M$ will contain the edge $(i - j)$. Finally, we specify the set of edges in the mixture graph $\mathcal{G}_M$ that are separable in each individual distribution $p_1$ and $p_2$ but become inseparable in the mixture distribution $p_M$.

**Definition 8 (Emergent Edge)** *The edge $(i - j)$ in the mixture graph $\mathcal{G}_M$ is called an emergent edge if the pair of nodes $i$ and $j$ are not adjacent in $\mathcal{G}_1$ and $\mathcal{G}_2$ but become inseparable in $p_M$. The set of emergent edges is given by*

$$\mathbf{E}_E \triangleq \{(i - j) : i, j \in \mathbf{V}, \ (i - j) \notin \mathbf{E}_U \ \wedge \ \nexists A \subseteq \mathbf{V} \setminus \{i, j\} : \ X_i \perp\!\!\!\perp X_j \mid X_A\}. \tag{8}$$

Based on the definitions of $\mathbf{E}_U$ and $\mathbf{E}_E$ we clearly have $\mathbf{E}_M = \mathbf{E}_U \cup \mathbf{E}_E$. Finally, we define a set of causal paths that pass through one or more nodes in $\Delta$. These causal paths are instrumental in specifying the separability conditions.

**Definition 9 ($\Delta$-through Path)** *We say that a causal path in $\mathcal{G}_\ell$ between $i$ and $j$ is a $\Delta$-through path, denoted by $i \overset{\Delta_\ell}{\rightsquigarrow} j$, if it passes through at least one node in $\Delta$, i.e., there exists $u \in \Delta$ such that $i \overset{\ell}{\rightsquigarrow} u \overset{\ell}{\rightsquigarrow} j$.*

## 3   Causal Discovery Objective

We aim to address the following questions. *What causal relationships can be learned about a mixture of DAGs? What is a graphical representation that can effectively specify these relationships?*

These questions are well-investigated when analyzing individual DAGs. In the individual DAG $\mathcal{G}_\ell$, each edge denotes an inseparable pair in $p_\ell$. The components of $\mathcal{G}_\ell$ that are learnable from observational data are characterized by a Markov equivalence class (Verma & Pearl, 1992) and represented by a completed partially DAG (CPDAG). Such notions and decisions do not have well-established counterparts when the observations are generated by a mixture distribution $p_M$ arising from a mixture of DAGs.

Identifying the inseparable node pairs is the cornerstone of constraint-based causal discovery. Therefore, the leap from the causal discovery of individual DAGs to causal discovery from a mixture of them critically hinges on determining the separability of nodes in the mixture model. We use the undirected mixture graph $\mathcal{G}_M$, which represents the inseparable pairs, to identify such pairs. This will be the critical initial step for the graphical representation of a mixture of DAGs. We note that the existence of emergent edges, i.e., the pair of nodes that are inseparable in mixture distribution but are separable in component distributions, is the main challenge that makes the causal discovery of a mixture of DAGs more complex than that of a single DAG. Therefore, our objectives in the paper are centered around separability analysis, and they include:

Objective$_1$: establishing the conditions that make a pair of nodes an emergent edge;

Objective$_2$: graphically representing these conditions by orienting the edges of the mixture graph;

Objective$_3$: and partially recovering these orientations.

# 4 Separability Analysis in Mixture Graphs

In this section, we first study Objective$_1$ and investigate the conditions under which a pair of nodes in a mixture model are inseparable. We start by considering an arbitrary pair of DAGs $\mathcal{G}_1$ and $\mathcal{G}_2$ without any constraints. We establish sufficient conditions under which any arbitrary pair of nodes in the mixture of $\mathcal{G}_1$ and $\mathcal{G}_2$ is separable. In the next step, we focus on tree-structured DAGs for which we tighten our sufficient conditions results and also show that these conditions are the necessary conditions, establishing the tightness of the results for tree-structured DAGs. For Objective$_2$, i.e., orienting the mixture graph, we introduce new edge notations and graphically represent the established necessary and sufficient conditions. For Objective$_3$, we analyze which of these edges can be inferred by performing conditional independence tests. For each result, we discuss the implications of our results and representations with respect to the existing representations of DAGs' mixtures. Finally, we devise an algorithm for learning the mixture graph based on our theoretical results. Proofs of the results are deferred to Appendix A.

## 4.1 Separability in General Graphs

Our separability analysis requires establishing a connection between the mixture distribution and the structure of the component DAGs. Composite DAG representation provides us with this connection through the following result.

**Theorem 1 (Markov property,(Saeed et al., 2020, Theorem 3.2))** *Let $A, B, C \subseteq \mathbf{V}$ be disjoint. If $\bar{A}$ and $\bar{B}$ are d-separated given $\bar{C}$ in composite DAG, then $X_A$ and $X_B$ are conditionally independent given $X_C$ in mixture distribution.*

We note that Theorem 1 in conjunction with the mixture faithfulness assumption implies that all conditional independencies that can be tested from $p_{\mathrm{M}}$ can be inferred by the separation statements in $\mathcal{G}_{\mathrm{C}}$. Using Theorem 1 and mixture faithfulness, we can immediately conclude that the definition of emergent edges in (8) is equivalent to

$$\mathbf{E}_{\mathrm{E}} = \{(i-j) : i, j \in \mathbf{V}, \ (i-j) \notin \mathbf{E}_{\mathrm{U}} \ \wedge \ \nexists A \subseteq \mathbf{V} \setminus \{i,j\} : \ i \perp\!\!\!\perp^{\mathrm{M}} j \mid A\} . \tag{9}$$

Hence, the separability of two nodes in mixture distribution is equivalent to the separability of the nodes in composite DAG $\mathcal{G}_{\mathrm{C}}$. Hence, an important observation is that the separability of two nodes depends on whether they have distinct conditional probability models in the component DAGs, i.e., their membership in $\Delta$ defined in (3). This is because the paths that stretch over $\mathcal{G}_1$ and $\mathcal{G}_2$ in $\mathcal{G}_{\mathrm{C}}$ necessarily pass through nodes in $\Delta$. The following theorem investigates all possible configurations and presents sufficient conditions for two nodes to be separable in a mixture of arbitrary DAGs.

**Theorem 2 (Separability – Sufficient Conditions)** *Consider nodes $i, j \in \mathbf{V}$ such that $i$ and $j$ are not adjacent in either component DAG, i.e., $(i-j) \notin \mathbf{E}_{\mathrm{U}}$.*

**Case 1)** $i \in \Delta, j \in \Delta$: *$i$ and $j$ are always inseparable.*

**Case 2)** $i \notin \Delta, j \notin \Delta$: *$i$ and $j$ are separable if the component DAGs do not have $\Delta$-through paths between $i$ and $j$ in opposite directions. That is, if one DAG contains a $\Delta$-through path from $i$ to $j$, then the other one does not have a $\Delta$-through path from $j$ to $i$.*

**Case 3)** $i \notin \Delta, j \in \Delta$: *$i$ and $j$ are separable if neither of the component DAGs contains a $\Delta$-through path from $i$ to $j$.*

The inseparability of the nodes $i, j \in \Delta$ is due to the path $i \overset{\ell}{\leftarrow} y \overset{\ell}{\rightarrow} j$ in composite DAG. The second case of Theorem 2 implies that if nodes $i \notin \Delta, j \notin \Delta$ are *inseparable*, i.e., they form an emergent edge, then for some distinct $\ell, \ell' \in \{1, 2\}$, we have $i \overset{\Delta_\ell}{\rightsquigarrow} j$ and $j \overset{\Delta_{\ell'}}{\rightsquigarrow} j$. The third case implies that if $i \notin \Delta$ and $j \in \Delta$ are inseparable, then we have $i \overset{\Delta_\ell}{\rightsquigarrow} j$ for some $\ell \in \{1, 2\}$. We have $|\Delta| \geq 2$ for all three cases. Hence, when $|\Delta| = 1$, the mixture of DAGs does not create any cycle, and we have $\mathbf{E}_{\mathrm{E}} = \emptyset$. The next observation of Theorem 2 is on the different formations of cycles in a mixture of DAGs. To highlight the shortcomings of the fused graph $\mathcal{G}_{\mathrm{F}}$ in representing a mixture of DAGs, we provide the following remark and example.

**Remark 1** *Note that $i \overset{1}{\rightsquigarrow} j$ and $j \overset{2}{\rightsquigarrow} i$ imply a cyclic relationship between $i$ and $j$ in fused graph $\mathcal{G}_{\mathrm{F}}$. However, the reverse direction is not always true, i.e., $i$ and $j$ can be part of a cycle in $\mathcal{G}_{\mathrm{F}}$ without being ancestors of each other in the component DAGs.*

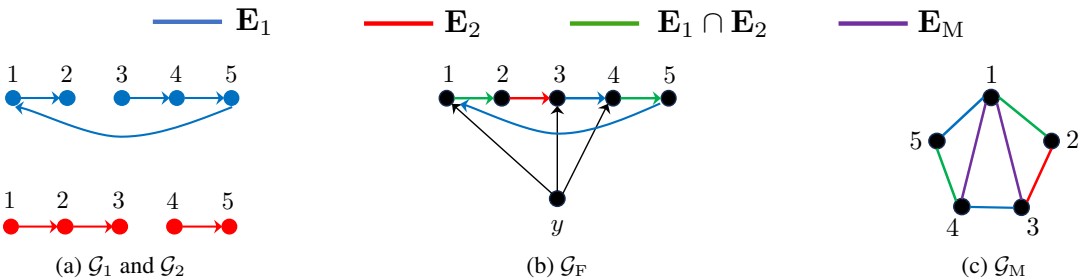

(a) $\mathcal{G}_1$ and $\mathcal{G}_2$         (b) $\mathcal{G}_F$         (c) $\mathcal{G}_M$

Figure 2: An example of the cases discussed in Remark 1. In this example, we have a cycle passing through all nodes. However, 2 and 4 are separable since there is no $\Delta$-through path from 2 to 4.

**Example.** Figure 2 illustrates the shortcomings of the fused graph through an example. Fused graph $\mathcal{G}_F$ allows us to visualize cycles in a single directed graph over $\mathbf{V} \cup \{y\}$. However, two nodes that are part of a cycle in this representation do not necessarily form an emergent edge. For instance, despite the cycle $(1 \to 2 \to 3 \to 4 \to 5 \to 1)$ in $\mathcal{G}_F$, neither of the pairs $(2 - 4)$ or $(3 - 5)$ forms an emergent edge. Furthermore, $\mathcal{G}_F$ implies that one of the paths $2 \to 3 \to 4$ (if 3 is not conditioned on) and $2 \to 3 \leftarrow y \to 4$ (if 3 is conditioned on) is always active. However, we have $2 \perp\!\!\!\perp^M 4 \mid 1$. In fact, 2 and 4 are unconditionally independent, showing that $\mathcal{G}_F$ is not a great representation. Therefore, for modeling a mixture of DAGs, it is important to distinguish between the directed paths in a component DAG and those formed by superposing the two component DAGs.

So far, we have discussed a mixture of DAGs without any structural constraints on the components of the mixture. Next, we focus on the mixtures of tree-structured DAGs and further improve the separability results.

## 4.2 Properties of Tree Mixtures

Imposing tree structures on DAGs $\mathcal{G}_1$ and $\mathcal{G}_2$ facilitates a tighter analysis of the separability of two nodes (or, equivalently, the conditions in which emergent edges are formed). The following theorem establishes sufficient conditions for the separability of a node pair in a mixture of tree DAGs. These sufficient conditions are less stringent than those presented for general DAGs in Theorem 2. Specifically, instead of ruling out all $\Delta$-through paths from $i$ to $j$, we rule out only a subset of them in which the child of $i$ on the path is in $\Delta$. Furthermore, we show that the refined conditions are necessary conditions too.

**Theorem 3 (Separability in Tree Structures – Necessary and Sufficient Conditions)** *Suppose that $\mathcal{G}_1$ and $\mathcal{G}_2$ are tree-structured DAGs. Consider nodes $i, j \in \mathbf{V}$ such that $i$ and $j$ are not adjacent in component DAGs, i.e., $(i-j) \notin \mathbf{E}_U$.*

**Case 1)** $i \in \Delta$ *and* $j \in \Delta$*: $i$ and $j$ are always inseparable.*
**Case 2)** $i \notin \Delta$ *and* $j \notin \Delta$*: $i$ and $j$ are separable if and only if component DAGs do not have $\Delta$-through paths between $i$ and $j$ in opposite directions such that the children of $i$ and $j$ on the paths are in $\Delta$.*
**Case 3)** $i \notin \Delta$ *and* $j \in \Delta$*: $i$ and $j$ are separable if and only if neither of the component DAGs contains a $\Delta$-through path from $i$ to $j$ such that the child of $i$ on the path is in $\Delta$.*

The first case is identical to the first case of Theorem 2. However, the sufficient conditions for $i$ and $j$ being separable in the second and third cases are tighter than their counterparts in Theorem 2. Specifically, the existence of $\Delta$-through paths is refined to the existence of $\Delta$-through paths with the additional condition that the child of the ancestor node of the path is in $\Delta$. Furthermore, this tightened sufficient condition is also shown to be necessary.

Theorem 3 also implies the necessary and sufficient conditions for $(i - j)$ being an emergent edge since if $(i - j) \notin \mathbf{E}_U$ and $i$ and $j$ are *inseparable*, then $(i - j)$ is an emergent edge. Formally, Case 2 implies that, for $i \notin \Delta, j \notin \Delta$, we have

$$(i-j) \in \mathbf{E}_E \iff \exists \ell, \ell' \in \{1,2\}, \ \exists u, v \in \Delta, u \neq v : i \xrightarrow{\ell} u \overset{\ell}{\rightsquigarrow} j, \ \text{and} \ j \xrightarrow{\ell'} v \overset{\ell'}{\rightsquigarrow} i. \tag{10}$$

Next, Case 3 implies that for $i \notin \Delta, j \in \Delta$, we have

$$(i-j) \in \mathbf{E}_E \iff \exists \ell \in \{1,2\}, \exists u \in \Delta : i \xrightarrow{\ell} u \overset{\ell}{\rightsquigarrow} j. \tag{11}$$

Hence, Theorem 3 provides a complete explanation for the emergent edges in mixture graph $\mathcal{G}_{\text{M}}$. Notably, the approach of Saeed et al. (2020) is not valid for a mixture of tree-structured DAGs in general. Specifically, motivated by the latent representation of the mixing variable via node $y$, (Saeed et al., 2020) constructs MAGs $\mathcal{M}_{\ell}$ from the sub-DAG induced by composite DAG $\mathcal{G}_{\text{C}}$ over the nodes $\mathcal{G}_{\ell} \cup \{y\}$ and takes a union of $\mathcal{M}_{\ell}$'s. It proceeds to learn the union MAG by performing fast causal inference (FCI) (Spirtes et al., 2000). However, to obtain a valid union MAG, $\mathcal{M}_1$ and $\mathcal{M}_2$ should have a shared partial order on $\mathbf{V}$. This assumption (called poset compatibility) is violated by the condition in (10) due to having a cycle in the mixture and by (11) due to having a causal path between two nodes in $\Delta$.

**Searching for a separating set.** After establishing the conditions for two nodes to be separable, our next result shows that the size of the smallest separating set is bounded.

**Lemma 1 (Size of the separating set for mixture of trees)** *Suppose that $\mathcal{G}_1$ and $\mathcal{G}_2$ are tree-structured DAGs $i$ and $j$ are separable in their mixture. Then, there exists a separating set $S$ with size at most $|S| \leq 3d$ where $d$ denotes the maximum in-degree of a node in any component DAG.*

We use Lemma 1 to establish the computational complexity of our approach in Section 4.3 and compare it to that of causal discovery in single DAGs.

**Interpreting the results from an interventional learning perspective.** By its definition, $\Delta$ can be viewed as the set of *soft intervention targets* for which $p_1$ and $p_2$ denote the observational and interventional distributions, respectively. As characterized in (Hauser & Bühlmann, 2012; Yang et al., 2018; Jaber et al., 2020) and further scrutinized in (Varıcı et al., 2021), additional information gained by observing samples from an interventional distribution amounts to inferring the non-intervened parents of an intervened node. This means the unoriented edges adjacent to a node $i \notin \Delta \cup \text{pa}(\Delta)$ in the CPDAG remain unoriented after the intervention. The parallel observation in our mixture model is as follows. If $i \notin \{\Delta \cup \text{pa}_1(\Delta) \cup \text{pa}_2(\Delta)\}$, then for any node $j$, the edge $(i - j)$ does not fit any of the conditions in Case 1 of Theorem 3, (10) and (11), implying that $(i - j) \notin \mathbf{E}_{\text{E}}$. Hence, node $i$ is unaffected by mixing $p_1$ and $p_2$, and the edges adjacent to node $i$ are due to component DAGs. Even though observing $p_{\text{M}}$ is strictly less informative than the setting of interventional causal learning literature where samples from $p_1$ and $p_2$ are observed separately, Theorem 3 reflects the importance of parents of $\Delta$ in a similar way.

**Orienting the edges of the mixture graph.** After establishing the conditions for two nodes to be separable and interpreting them, we look into Objective$_2$ of assigning orientations to the inseparable pairs, i.e., edges of $\mathcal{G}_{\text{M}}$, to represent these conditions. We introduce the following edge notations ($\twoheadrightarrow$) and ($\twoheadleftarrow\!\!\twoheadrightarrow$) for orienting the undirected mixture edges $\mathbf{E}_{\text{M}}$.

**Edge $i \twoheadrightarrow j$:** We use this notation to refer to mixture edges in the following cases.

- The edges that point towards $j \notin \Delta$ in both graphs.
- Emergent edges that point towards $j \in \Delta$.

**Edge $i \twoheadleftarrow\!\!\twoheadrightarrow j$:** We use this notation to refer to the mixture edges in the following cases.

- Emergent edges between $i \notin \Delta$ and $j \notin \Delta$.
- Mixture edges between $i \in \Delta$ and $j \in \Delta$.

We orient the mixture edges $\mathbf{E}_{\text{M}}$ according to these edge notations and denote the oriented edges by $\mathbf{E}_{\text{M}}^*$. Finally, we define $\mathcal{G}_{\text{M}}^* \triangleq (\mathbf{V}, \mathbf{E}_{\text{M}}^*)$ to denote the oriented version of the mixture graph $\mathcal{G}_{\text{M}}$. Figure 3 illustrates an example of constructing $\mathcal{G}_{\text{M}}^*$.

**Inferring orientations.** Finally, we address Objective$_3$ by partially recovering edge orientations. To that end, we investigate what can be inferred about $\mathbf{E}_{\text{M}}^*$ via a CI oracle. By its definition in (7), undirected edges $\mathbf{E}_{\text{M}}$ can be identified by exhaustively performing CI tests. The definition of a collider carries to our representation, i.e., node $k$ is a collider on a path if edges on both sides of $k$ have an arrow pointing towards $k$. Analogous to the (partial) recovery of the edge orientations of a single DAG, we rely on the separability conditions of unshielded triples of $(i, k, j)$ in $\mathcal{G}_{\text{M}}$. The following lemma summarizes the partial information we can recover about $\mathbf{E}_{\text{M}}^*$.

**Lemma 2** *Suppose that $\mathcal{G}_1$ and $\mathcal{G}_2$ are tree-structured DAGs. Consider nodes $i \notin \Delta$ and $j \in \mathbf{V}$ that are separated by set $S$. Consider node $k \in \mathbf{V} \setminus S$ for which $(i, k, j)$ is an unshielded triples in $\mathcal{G}_{\text{M}}$. Then, we have the following edge orientations in $\mathbf{E}_{\text{M}}^*$.*

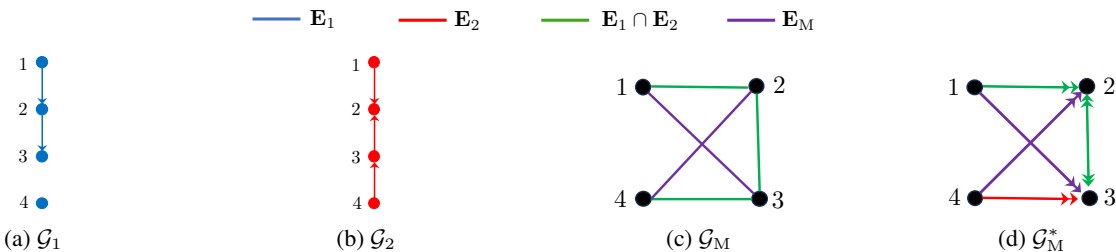

Figure 3: An example of (a)-(b) Component DAGs; (c) associated undirected mixture graph; (d) oriented mixture graph.

1. *If $j \notin \Delta$, $k \in \Delta$, then we have $i \twoheadrightarrow k \twoheadleftarrow j$.*

2. *If $j \in \Delta$, $k \in \Delta$, then we have $i \twoheadrightarrow k \twoheadleftrightarrow j$.*

3. *If $j \notin \Delta$, $k \notin \Delta$, then we have $i \twoheadrightarrow k \twoheadleftarrow j$ or $i \twoheadleftrightarrow k \twoheadleftrightarrow j$.*

4. *If $j \in \Delta$, $k \notin \Delta$, then we have $i \twoheadrightarrow k \twoheadleftarrow j$ or $i \twoheadleftrightarrow k \twoheadleftarrow j$ or $i \twoheadleftrightarrow k \twoheadrightarrow j$.*

**Connections to v-structures.** Strobl (2023) has provided an example of a DAGs' mixture with additional latent nodes where v-structures cannot be identified from a CI oracle alone. We make a similar observation in Case 4 of Lemma 2 that some v-structures in the mixture cannot be inferred without additional information (for instance, $i \twoheadleftrightarrow k \twoheadleftarrow j$ and $i \twoheadleftrightarrow k \twoheadrightarrow j$ are indistinguishable). On the other hand, Case 1 and 2 of Lemma 2 show that if we know $\Delta$, we can orient edges of an unshielded triple upon $k \in \Delta$. Hence, (partial) knowledge of the set $\Delta$, which may be available in certain applications, allows us to infer important information about the cause-effect relationships in a DAGs' mixture.

### 4.3 Algorithmic Perspective

We leverage our theoretical results to devise an algorithm for learning the mixture graph. The details of the algorithm are as follows. Algorithm 1 takes samples from the mixture distribution $p_M$ as the main input and $\hat{\Delta} \subseteq \Delta$ as an *optional* input when it is available, which accounts for the (possibly partial) knowledge of the nodes with varying conditional distributions. We assume access to a CI oracle to perform conditional independence tests. The algorithm has two main stages:

**Stage 1:** In Stage 1, we start with a complete graph, then consider every node pair and remove the edge between them if there exists a conditioning set that makes the random variables corresponding to the node pair independent. We record the separating sets for each removed edge to use in the orientation phase. Note that this skeleton learning phase applies to a mixture of arbitrary graphs.

**Stage 2:** If the component DAGs are known to have tree structures, we continue with Stage 2 and use the additional input $\hat{\Delta}$ to partially orient the skeleton of the mixture graph. As discussed in Section 4, set $\Delta$ plays a crucial role in the separability of nodes $i, j \in \mathbf{V}$ in mixture distribution and the orientation of the inseparable pairs $(i - j) \in \mathbf{E}_M$. Therefore, we first orient the edges between $i \in \hat{\Delta}$ and $j \in \hat{\Delta}$ nodes. Next, we consider the unshielded triples in the skeleton such that the connecting node of the triple is not contained in the separating set of the non-adjacent nodes of the triple. While orienting the edges, a circle on one end denotes the ambiguity about the orientation, e.g., we use $\circ\!\!\twoheadrightarrow$ to denote that the edge can be either $\twoheadrightarrow$ or $\twoheadleftarrow$. We follow this convention to orient as many edges as possible. If we have partial knowledge of $\Delta$, e.g., we are given $\hat{\Delta} \subseteq \Delta$, we only orient one end of the edges that point towards nodes in $\Delta$ in an unshielded triple. However, if we know that given $\hat{\Delta}$ is exactly equal to $\Delta$, we can orient more edges by considering each unshielded triple with respect to its nodes' inclusion in $\Delta$.

In the rest of this section, we connect the outputs of Algorithm 1 to the theoretical results presented before. The following lemma states the implications of an edge in the output of Stage 1 and is essentially equivalent to Theorem 2.

**Lemma 3 (Stage 1 Output)** *Assume that the mixture faithfulness assumption holds. If $(i - j) \in \mathbf{E}_M$ at the end of Stage 1 of Algorithm 1, then there is an edge between $i$ and $j$ in at least one of the component DAGs, and otherwise one of the following statements is true:*

---

**Algorithm 1**

---

1: **Input:** Samples from mixture distribution $p_{\mathrm{M}}$, $\hat{\Delta}$

---

2: **Stage 1: Skeleton of $\mathcal{G}_{\mathrm{M}} = (\mathbf{V}, \mathbf{E}_{\mathrm{M}})$**
3: Form complete undirected graph: $\mathbf{E}_{\mathrm{M}} \leftarrow \{(i-j) : i \in \mathbf{V}, j \in \mathbf{V}\}$
4: **for** all $i, j \in \mathbf{V}$ **do**
5:     **for** all $S \in \mathbf{V} \setminus \{i, j\}$ **do**                           ▷ for trees, only for the sets: $|S| \leq 3d$
6:         **if** $X_i \perp\!\!\!\perp X_j \mid X_S$ **then**
7:             Remove $(i-j)$ edge: $\mathbf{E}_{\mathrm{M}} \leftarrow \mathbf{E}_{\mathrm{M}} \setminus (i-j)$, and $\mathrm{SepSet}(i,j) \leftarrow S$.
8:             **break**
9:         **end if**
10:     **end for**
11: **end for**

---

12: **Stage 2: Orientation rules for a mixture of trees**
13: Denote all pairs $(i-j) \in \mathbf{E}_{\mathrm{M}}$ as $i \circ\!\!-\!\!\circ j$: $\tilde{\mathbf{E}}_{\mathrm{M}} \leftarrow \{i \circ\!\!-\!\!\circ j : (i-j) \in \mathbf{E}_{\mathrm{M}}\}$
14: **for** all $i, j \in \hat{\Delta}$ **do**
15:     Orient $i \leftarrow\!\!\!\leftarrow\!\!\!\rightarrow\!\!\!\rightarrow j$ in $\tilde{\mathbf{E}}_{\mathrm{M}}$
16: **end for**
17: Unshielded triples with non-separator connectors:

$$\mathcal{U} \leftarrow \{(i, k, j) : (i-k) \in \mathbf{E}_{\mathrm{M}}, (j-k) \in \mathbf{E}_{\mathrm{M}}, (i-j) \notin \mathbf{E}_{\mathrm{M}},\ k \notin \mathrm{SepSet}(i,j)\}$$

18: **if** we have $\hat{\Delta} = \Delta$ **then**
19:     **for** all $(i, k, j) \in \mathcal{U}$ **do**
20:         **if** $k \in \Delta, i \notin \Delta, j \notin \Delta$ **then**
21:             Orient $i \rightarrow\!\!\!\rightarrow k \leftarrow\!\!\!\leftarrow j$ in $\tilde{\mathbf{E}}_{\mathrm{M}}$
22:         **else if** $k \in \Delta, i \notin \Delta, j \in \Delta$ **then**
23:             Orient $i \rightarrow\!\!\!\rightarrow k \leftarrow\!\!\!\leftarrow\!\!\!\rightarrow\!\!\!\rightarrow j$ in $\tilde{\mathbf{E}}_{\mathrm{M}}$
24:         **else if** $k \in \Delta, i \in \Delta, j \notin \Delta$ **then**
25:             Orient $i \leftarrow\!\!\!\leftarrow\!\!\!\rightarrow\!\!\!\rightarrow k \leftarrow\!\!\!\leftarrow j$ in $\tilde{\mathbf{E}}_{\mathrm{M}}$
26:         **else if** $k \notin \Delta, i \notin \Delta, j \notin \Delta$ **then**
27:             Orient $i \circ\!\!\rightarrow\!\!\!\rightarrow k \leftarrow\!\!\!\leftarrow\!\!\circ j$ in $\tilde{\mathbf{E}}_{\mathrm{M}}$
28:         **else if** $k \notin \Delta, i \notin \Delta, j \in \Delta$ **then**
29:             Orient $i \circ\!\!\rightarrow\!\!\!\rightarrow k \circ\!\!-\!\!\circ j$ in $\tilde{\mathbf{E}}_{\mathrm{M}}$
30:         **else if** $k \notin \Delta, i \in \Delta, j \notin \Delta$ **then**
31:             Orient $i \circ\!\!-\!\!\circ k \leftarrow\!\!\!\leftarrow\!\!\circ j$ in $\tilde{\mathbf{E}}_{\mathrm{M}}$
32:         **end if**
33:     **end for**
34: **else if** we have $\hat{\Delta} \subset \Delta$ **then**
35:     **for** all $(i, k, j) \in \mathcal{U}$ **do**
36:         **if** $k \in \hat{\Delta}$ **then**
37:             Orient $i \circ\!\!\rightarrow\!\!\!\rightarrow k \leftarrow\!\!\!\leftarrow\!\!\circ j$ in $\tilde{\mathbf{E}}_{\mathrm{M}}$
38:         **end if**
39:     **end for**
40: **end if**
41: **return** $\mathbf{E}_{\mathrm{M}}, \tilde{\mathbf{E}}_{\mathrm{M}}$

---

*1. $i \in \Delta$ and $j \in \Delta$.*

*2. $i \notin \Delta$ and $j \notin \Delta$, and component DAGs have $\Delta$-through paths between $i$ and $j$ in opposite directions.*

*3. $i \notin \Delta$ and $j \in \Delta$, and at least one of the component DAGs contains a $\Delta$-through path from $i$ to $j$.*

Next, we provide the separability necessary and sufficient conditions for the output of Stage 1 when the component DAGs have tree structures. These conditions are equivalent to those for separability given in Theorem 3.

**Lemma 4 (Separability in Tree Structures)** *Suppose that $\mathcal{G}_1$ and $\mathcal{G}_2$ are tree-structured DAGs. Consider the skeleton $\mathbf{E}_{\mathrm{M}}$ at the end of Stage 1 of Algorithm 1 and assume that mixture faithfulness holds. If there is an edge between $i$ and $j$ in at least one of the component DAGs, then $(i - j) \in \mathbf{E}_{\mathrm{M}}$. Otherwise, we have*

**Case 1)** $i \in \Delta$ *and* $j \in \Delta$: $(i - j)$ *is in* $\mathbf{E}_{\mathrm{M}}$.

**Case 2)** $i \notin \Delta$ *and* $j \notin \Delta$: $(i - j)$ *is in* $\mathbf{E}_{\mathrm{M}}$ *if and only if component DAGs have $\Delta$-through paths between $i$ and $j$ in opposite directions such that the children of $i$ and $j$ on the paths are in $\Delta$.*

**Case 3)** $i \notin \Delta$ *and* $j \in \Delta$: $(i - j)$ *is in* $\mathbf{E}_{\mathrm{M}}$ *if and only if at least one component DAG has a $\Delta$-through path from $i$ to $j$ such that the child of $i$ on the path is in $\Delta$.*

Finally, we consider the edge orientations in Stage 2. In Section 4.2, we defined edge orientations $\{\twoheadrightarrow, \leftarrow\!\!\twoheadrightarrow\}$ and constructed oriented mixture graph $\mathcal{G}_{\mathrm{M}}^* = (\mathbf{V}, \mathbf{E}_{\mathrm{M}}^*)$ accordingly. Then, in Lemma 2, we have established what partial orientations can be learned from unshielded triples. Under mixture faithfulness, Stage 2 of Algorithm 1 follows Lemma 2 to construct partially oriented edge set $\tilde{\mathbf{E}}_{\mathrm{M}}$.

**Computational complexity** We note that for general graphs, Theorem 2 provides only sufficient conditions for the separability of two nodes without establishing matching worst-case necessary conditions. Therefore, for general graphs, Stage 1 has $O(n^2 \cdot 2^n)$ exponential complexity. On the contrary, for a mixture of trees, Lemma 1 shows that checking all sets $S$ of size at most $3d$ is sufficient to identify all mixture edges. This procedure is repeated for $\binom{n}{2}$ node pairs. Note that Stage 2 directly reads off the unshielded triples and saved separating sets for all separable node pairs from Stage 1 output. Therefore, the total complexity of Algorithm 1 for a mixture of tree-structured DAGs is $\mathcal{O}(n^{3d+2})$. We also recall that the complexity of the well-known PC algorithm (Spirtes et al., 2000) for structure learning of a general DAG is $\mathcal{O}(n^{d+2})$. The difficulty in our setting comes from having to block all paths between nodes $i$ and $j$ across multiple graphs, which may contain causal relationships in opposite directions.

## 5 Empirical Evaluations

We evaluate the performance of Algorithm 1 for estimating the skeleton of the mixture graph and partially orienting the recovered edges using synthetic data.

**Experimental setup.** To generate $\mathcal{G}_1$ and $\mathcal{G}_2$, we use Erdős-Rényi model with $G(n, p)$ with density $p = 2/n$ for $n \in \{6, 8\}$ for each of the component DAGs. We repeat sampling graphs from Erdős-Rényi model until both $\mathcal{G}_1$ and $\mathcal{G}_2$ have tree structures. Note that the component DAGs are generated independently and do not necessarily share the same topological order. For the causal relationships, we follow linear structural equation models (SEM) with additive Gaussian noise for both component DAGs. Edge weights are sampled uniformly in $\pm[0.25, 2]$, and shared directed edges between two graphs are assigned the same weights. The mean of the Gaussian noise for each node is sampled uniformly in $[-2, 2]$ with a standard deviation of 1 and set to be equal in both graphs. Note that this parameterization implies that $\Delta$ consists of the nodes with varying parent sets across two graphs. We use a partial correlation test as a conditional independence test routine for simplicity even though the true distribution $p_{\mathrm{M}}$ is a mixture of Gaussian distributions. The threshold for the $p$-value of the CI test is set to $\alpha = 0.1$. For each case, we run the algorithm with $s \in \{1e3, 3e3, 1e4, 3e4, 5e4, 1e5\}$ number of samples from each component DAG. We repeat this experimental procedure for 500 randomly generated $\mathcal{G}_1$ and $\mathcal{G}_2$ tree pairs and report the average results. We investigate the effects of the choice of CI test and varying mixing rate of two component distributions in Appendix C.

**Skeleton recovery.** Figure 4 illustrates the precision and recall rates for recovering the skeleton $\mathbf{E}_{\mathrm{M}}$ with varying numbers of samples for $n = 6$ and $n = 8$. The observations are shared between two graph sizes, with the performance at smaller graphs being higher, as expected. The first observation is that we have very high precision rates even with as few as $s = 1000$ samples (shown by the blue curves). On the other hand, high recall rates become possible given a large enough number of samples (shown by the orange curves). Recall that mixture edges consist of two disjoint sets: edges that exist in at least one component DAG (i.e., union edges) and emergent edges. We further look into the decomposition of the overall recall rate into the recall rate of union and emergent edges (green and red curves,

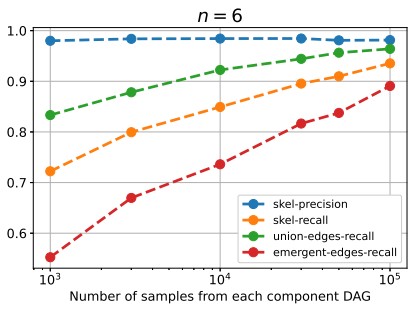
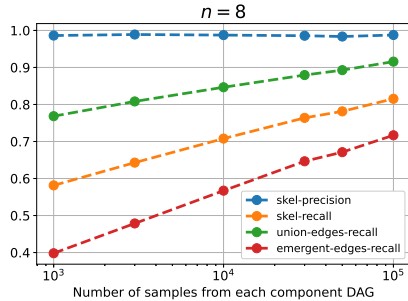

(a) Skeleton recovery results for $n = 6$

(b) Skeleton recovery results for $n = 8$

Figure 4: Skeleton recovery rates

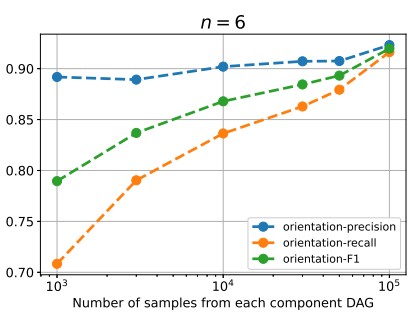
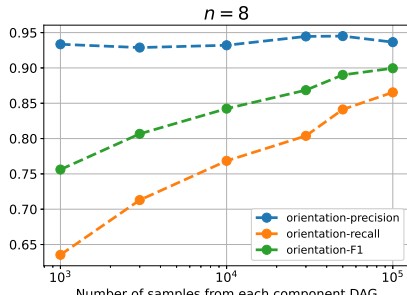

(a) Orientation recovery results for $n = 6$

(b) Orientation recovery results for $n = 8$

Figure 5: Orientation recovery rates

respectively). The main observation is that emergent edges require more samples than union edges to achieve the same recall rates. This can be explained by the dependence of the two nodes of an emergent edge is only due to the mixing of two distributions. In contrast, a union edge denotes a strong dependence between two nodes in at least one component DAG, which comprises half of the observed samples. It is, however, noteworthy that the gap between the recall rates of union and emergent edges narrows as the number of samples increases.

**Orientation recovery.** For recovering the orientation of the edges, we take $\hat{\Delta} = \Delta$ as input for evaluating our results for diverse combinations of node pairs. In this case, $(i \leftrightarrow\!\!\!\rightarrow j)$ orientations for $i \in \Delta$ and $j \in \Delta$ are already known by the input $\Delta$. Hence, we report the results for the recovery of non-trivial orientations of $\rightarrow\!\!\!\rightarrow$ and $\circ\!\!\rightarrow\!\!\!\rightarrow$. Figure 5 plots the precision, recall, and F1 rates for recovering the oriented edges $\tilde{\mathbf{E}}_M$ with varying numbers of samples for $n = 6$ and $n = 8$. The first observation is similar to skeleton recovery in the sense that precision rates are higher than recall. We note that emergent edges had relatively lower recall rates in Figure 4. Excluding the emergent edges between two nodes in $\Delta$ for orientation recovery (due to already being oriented given $\Delta$) explains the higher recall rates in Figure 5 compared to Figure 4. We also see that precision rates in Figure 5 are slightly lower than the ones in Figure 4. This is due to the fact that separating set decisions in Stage 1 of Algorithm 1 may contain errors even if the undirected edge of the skeleton is correctly recovered.

## 6 Discussion

**Summary.** In this paper, we have taken the challenge of causal discovery in a mixture of DAGs. We focused on analyzing the key node pairs that are inseparable in the mixture model but separable in individual DAGs. In our analysis, we have provided sufficient conditions for separating a pair of nodes for a mixture of arbitrary DAGs. We have further improved these conditions and have shown them to be necessary for a mixture of tree-structured DAGs. We have also formalized a novel graphical characterization to represent the new causal relationships that arise in the DAGs' mixture and established results for partially recovering the orientations of the graph's edges.

**Limitations.** The main limitation of this paper is some of our results (Section 4.2) are exclusive to a mixture of tree-structured DAGs. A practical limitation can be the computational burden of checking every possible separating set to determine if two nodes are inseparable in the mixture.

**Future work.** This paper formalizes the separability framework for the broad problem of causal discovery of a mixture of DAGs. Extending the theoretical results and graphical representations for a mixture of trees to a mixture of arbitrary graphs is an important future direction. Developing efficient algorithms for identifying the inseparable node pairs (similar to the PC algorithm for structure learning of a single DAG) is important to accommodate a mixture of DAGs with a large number of nodes in practice. Finally, generalizing the mixture data model to a block mixture model can provide essential additional information for the causal discovery of the mixture graph. For instance, in a dynamical system in which there is a gradual, smooth transition between the component models, one can perform change-point detection and dynamic model tracking and estimating. We note that such a setting would also require developing a counterpart for the graphical models for the new mixture distribution while still preserving Markov properties.

## Acknowledgements

This work was supported by the Rensselaer-IBM AI Research Collaboration (http://airc.rpi.edu), part of the IBM AI Horizons Network (http://ibm.biz/AIHorizons).

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

# A Proofs of the Main Results

## A.1 Invariant conditional distributions and parent sets

The following result states that if a node $i$ has invariant causal mechanisms across two DAGs, i.e., invariant distributions conditioned on its respective parents in $\mathcal{G}_1$ and $\mathcal{G}_2$, then the parent sets must be equal.

**Lemma 5** *If $i \notin \Delta$, then we have $\mathrm{pa}_1(i) = \mathrm{pa}_2(i)$.*

**Proof** Consider a node $i \notin \Delta$. We prove that $\mathrm{pa}_1(i) = \mathrm{pa}_2(i)$ by contradiction. Suppose the contrary and assume $\mathrm{pa}_1(i) \neq \mathrm{pa}_2(i)$. Without loss of generality, let $\mathrm{pa}_2(i) \setminus \mathrm{pa}_1(i) \neq \emptyset$, and consider $j \in \mathrm{pa}_2(i) \setminus \mathrm{pa}_1(i)$. Let $S \triangleq \mathrm{pa}_2(i) \setminus \{j\}$. Since $i \notin \Delta$, the conditional probability distributions of $i$ given its parents are the same, i.e.,

$$p_1(x_i \mid x_{\mathrm{pa}_1(i)}) = p_2(x_i \mid x_{\mathrm{pa}_2(i)}) = p_2(x_i \mid x_S, x_j) . \tag{12}$$

By definition of a causal parent, there exist realizations $X_S = \theta^*$, $X_j = a^*$ and $X_j = b^*$ such that

$$p_2(X_i = x_i \mid X_S = \theta^*, X_j = a^*) \neq p_2(X_i = x_i \mid X_S = \theta^*, X_j = b^*) . \tag{13}$$

Otherwise, $X_j$ has no effect on $X_i$ conditioned on set $S$, and it is not a parent of $i$ in $\mathcal{G}_2$. Now consider a realization of $X_{\mathrm{pa}_1(i)} = \varphi^*$ for which the values of $X_{\mathrm{pa}_1(i) \cap \mathrm{pa}_2(i)}$ are determined by $\theta^*$. Then, we have

$$p_1(X_i = x_i \mid X_{\mathrm{pa}_1(i)} = \varphi^*) = p_2(X_i = x_i \mid X_S = \theta^*, X_j) . \tag{14}$$

Note that the left-hand side of (14) is independent of $X_j$. However, the right-hand side of (14) takes distinct values for realizations $X_j = a^*$ and $X_j = b^*$, which is a contradiction. Therefore, $\mathrm{pa}_2(i) \setminus \mathrm{pa}_1(i) = \emptyset$ and by similar line of argument, we have $\mathrm{pa}_1(i) \setminus \mathrm{pa}_2(i) = \emptyset$. This concludes the proof that if $i \notin \Delta$, then $\mathrm{pa}_1(i) = \mathrm{pa}_2(i)$.

## A.2 Proof of Theorem 2

First, we provide a counterpart of Theorem 2, which states the necessary conditions for nodes $i$ and $j$ being inseparable (forming an emergent edge). The proof of Theorem 2 will readily follow from this result.

**Theorem 4** *Consider nodes $i, j \in \mathbf{V}$ such that $i$ and $j$ are not adjacent in either component DAG, i.e., $(i - j) \notin \mathbf{E}_{\mathrm{U}}$.*

**Case 1)** *$i \in \Delta$ and $j \in \Delta$: $i$ and $j$ are always inseparable, i.e., $(i - j)$ is an emergent edge.*

**Case 2)** *$i \notin \Delta$ and $j \notin \Delta$: If $i$ and $j$ are inseparable, then the component DAGs have $\Delta$-through paths between $i$ and $j$ in opposite directions. That is, one DAG contains a $\Delta$-through path from $i$ to $j$ and the other one contains a $\Delta$-through path from $j$ to $i$.*

**Case 3)** *$i \notin \Delta$ and $j \in \Delta$: If $i$ and $j$ are inseparable, then at least one component DAG contains a $\Delta$-through path from $i$ to $j$.*

**Proof** The first case is trivial as inseparability of the nodes $i \in \Delta$ and $j \in \Delta$ is due to the path $i \overset{\ell}{\leftarrow} y \overset{\ell}{\rightarrow} j$ in composite DAG. We prove the non-trivial cases as follows.

- **Case 2**: For $i \notin \Delta$ and $j \notin \Delta$, we have $\mathrm{pa}_1(i) = \mathrm{pa}_2(i)$ and $\mathrm{pa}_1(j) = \mathrm{pa}_2(j)$ by Lemma 5. Let the pair $(i - j)$ be inseparable and consider the set $S = \mathrm{pa}(i) \cup \mathrm{pa}(j)$. Then, there exists an open path $\pi$ between $\bar{i}$ and $\bar{j}$ given $\bar{S}$ on composite DAG $\mathcal{G}_{\mathrm{C}}$. Since $\mathrm{pa}(i)$ is given, any path that starts with $i \overset{\ell}{\leftarrow}$ is blocked for $\ell \in \{1, 2\}$. Without loss of generality, let $\pi$ start with an edge $i \overset{1}{\rightarrow}$. We investigate two possibilities for $\pi$.

  - Let $y \notin \pi$. Then, $\pi$ must be of the form $i \overset{1}{\rightarrow} \cdots \overset{1}{\leftarrow} j$. Without loss of generality, suppose that $i \notin \mathrm{de}_1(j)$. However, there exists a collider $k$ on $\pi$ such that $k \in \mathrm{de}_1(j)$. Then, for $\pi$ to be open, a node $z \in \mathrm{de}_1^+(k)$, which is also a node in $\mathrm{de}_1(j)$, must be given in $S$. However, $z \notin \mathrm{pa}(i)$ since $i \notin \mathrm{de}_1(j)$ and $z \notin \mathrm{pa}(j)$ due to acyclicity. Therefore, the case of $y \notin \pi$ is not possible.

- Let $y \in \pi$. Without loss of generality, suppose that $\pi$ contains $i \overset{1}{\leadsto} k \overset{1}{\leftarrow} \ldots y$. For $\pi$ to be open, a $z \in \mathrm{de}_1^+(k)$ must be given in $S = \mathrm{pa}(i) \cup \mathrm{pa}(j)$, which implies that $k \in \mathrm{an}_1(j)$. Subsequently, $i \overset{1}{\leadsto} j$. Repeating the same line of arguments, we find that $j \overset{2}{\leadsto} i$. Finally, note that if none of the nodes on the path $i \overset{1}{\leadsto} j$ is not contained in $\Delta$, then we have the same path in $\mathcal{G}_2$ as $i \overset{2}{\leadsto} j$, which contradicts with $j \overset{2}{\leadsto} i$. Therefore, there exist $\Delta$-through paths between $i$ and $j$ in opposite directions in the two component DAGs.

- **Case 3**: Since $i \notin \Delta$, we have $\mathrm{pa}_1(i) = \mathrm{pa}_2(i)$. Let $(i-j)$ be inseparable and consider the set $S = \mathrm{pa}(i)$. Then, there exists an open path $\pi$ between $i$ and $j$ given $S$ on composite DAG $\mathcal{G}_\mathrm{C}$. Since $\mathrm{pa}(i)$ is given, any path that starts with $i \overset{\ell}{\leftarrow}$ is blocked, and $\pi$ necessarily starts with $i \overset{\ell}{\rightarrow}$ edge. If $\pi$ contains a collider $k$ such that $i \overset{\ell}{\leadsto} k$, then $S = \mathrm{pa}(i)$ must contain a node from $\mathrm{de}_\ell^+(k)$ for $\pi$ to be open. This contradicts with $k \in \mathrm{de}_\ell(i)$. Hence, there is no collider on $\pi$ and we have $i \overset{\ell}{\leadsto} j$. Since $j \in \Delta$, $i \overset{\ell}{\leadsto} j$ is a $\Delta$-through path.

**Proof of Theorem 2.** Case 1 of Theorem 2 is the same as Case 1 of Theorem 4. For a pair of nodes $i, j$ that are not adjacent in component DAGs, we define the following statements.

$$P : i \text{ and } j \text{ are separable },$$
$$Q : \text{component DAGs do not have } \Delta\text{-through paths between } i \text{ and } j \text{ in opposite directions}$$
$$R : \text{neither of the component DAGs contains a } \Delta\text{-through path from } i \text{ to } j$$

Then, Case 2 of Theorem 4 shows that if $i \notin \Delta$ and $j \notin \Delta$, we have $\neg P \implies \neg Q$. This equivalently means that $Q \implies P$, which is Case 2 of Theorem 2. Next, Case 3 of Theorem 4 shows that if $i \notin \Delta$ and $j \in \Delta$, we have $\neg P \implies \neg R$. This equivalently means that $R \implies P$, which is Case 3 of Theorem 2, and the proof is completed.

## A.3 Proof of Theorem 3

Case 1 is the same as Case 1 of Theorem 2 and is trivial since inseparability of the nodes $i \in \Delta$ and $j \in \Delta$ is due to the path $i \overset{\ell}{\leftarrow} y \overset{\ell}{\rightarrow} j$ in composite DAG. We prove the non-trivial cases as follows.

- **Case 2:** $i \notin \Delta$ **and** $j \notin \Delta$. Note that the theorem statement is equivalent to: *$i$ and $j$ are inseparable if and only if the component DAGs have $\Delta$-through paths between $i$ and $j$ in opposite directions such that the children of $i$ and $j$ on the paths are in $\Delta$.* We prove the two directions of this equivalent statement as follows.

  - Let there exist $\Delta$-through paths between $i$ and $j$ in opposite directions such that the children of $i$ and $j$ on the paths are in $\Delta$. Then, without loss of generality, let $i \overset{1}{\rightarrow} u \overset{1}{\leadsto} j$ and $j \overset{2}{\rightarrow} v \overset{2}{\leadsto} i$ where $u, v \in \Delta$. Suppose that $(i-j)$ is not an emergent edge, and set $S$ separates them. Then, $S$ must contain nodes from $\mathrm{de}_1^+(u)$ and $\mathrm{de}_2^+(v)$ to block both $\Delta$-through paths. However, this opens the path $i \overset{1}{\rightarrow} u \overset{1}{\leftarrow} y \overset{2}{\rightarrow} v \overset{2}{\leftarrow} j$, which contradicts with $i$ and $j$ being separated. Therefore, $i$ and $j$ are inseparable and $(i-j) \in \mathbf{E}_\mathrm{E}$.

  - Let $(i-j)$ be inseparable. By Theorem 4, without loss of generality, we have $i \overset{1}{\rightarrow} u \overset{1}{\leadsto} j$ and $j \overset{2}{\rightarrow} v \overset{2}{\leadsto} i$. We prove that $u \in \Delta$ by contradiction. Suppose the contrary and let $u \notin \Delta$. Consider $S = \mathrm{pa}(i) \cup \mathrm{pa}(j) \cup \mathrm{pa}^+(u) \cup \backslash \{i\}$. Let $\pi$ be an open path given $S$. Since $\mathrm{pa}(i)$ and $\mathrm{pa}(j)$ are given, $\pi$ starts with $i \overset{\ell}{\rightarrow}$ edge and is not a causal path from $i$ to $j$. Then, $\pi$ must contain a collider $k \in \mathrm{de}_\ell(i)$, i.e., $\pi$ contains $i \overset{\ell}{\leadsto} k \overset{\ell}{\leftarrow}$. For $\pi$ to be open, some $z \in \mathrm{de}_\ell(k)$ must be in $S$. Due to acyclicity, $z \notin \mathrm{pa}(i)$. Recall that $\mathcal{G}_\ell$ is a tree, and there can be only one directed path from $i$ to $j$ in $\mathcal{G}_\ell$. Then, if $z \in \mathrm{pa}(j)$, having $i \overset{\ell}{\rightarrow} u \overset{\ell}{\leadsto} j$ and $i \overset{\ell}{\leadsto} k \overset{\ell}{\leadsto} z \overset{\ell}{\rightarrow} j$ imply that $u$ is on $\pi$ and blocks $\pi$, which contradicts with $\pi$ being open. If $z \in \mathrm{pa}^+(u)$, then $i \overset{\ell}{\leadsto} z$ and $i \overset{\ell}{\rightarrow} u$ imply that $z = u$. In this case, $\pi$ becomes $i \overset{\ell}{\rightarrow} u \overset{\ell}{\leftarrow}$, which is blocked by $\mathrm{pa}(u)$, and $\pi$ is not open. We find a contradiction in each possible configuration for $u \notin \Delta$. Therefore, we find that $u \in \Delta$. By following similar line arguments, we can find that $v \in \Delta$ as well, which concludes the proof of this case.

- **Case 3:** $i \notin \Delta$ **and** $j \in \Delta$. Note that the theorem statement is equivalent to the following statement: *$i$ and $j$ are inseparable if and only then at least one component DAG contains a $\Delta$-through path from $i$ to $j$ such that the child of $i$ on the path is in $\Delta$.* We prove the two directions of this equivalent statement as follows.

- Let $i \xrightarrow{1} u \overset{1}{\rightsquigarrow} j$ without loss of generality such that $u \in \Delta$. Suppose that $(i - j)$ is not an emergent edge, and set $S$ separates them. Then, $S$ must contain a node from $\mathrm{de}_1^+(u)$ to block the causal path $i \overset{1}{\rightsquigarrow} j$. However, conditioning on a node from $\mathrm{de}_1^+(u)$ opens the path $i \xrightarrow{1} u \xleftarrow{1} y \xrightarrow{1} j$, which contradicts with $i$ and $j$ being separated by $S$. Therefore, $i$ and $j$ are inseparable.

- Let $(i - j)$ be inseparable. By Theorem 4, without loss of generality, we have $i \xrightarrow{1} u \overset{1}{\rightsquigarrow} j$. We prove that $u \in \Delta$ by contradiction. Suppose the contrary and let $u \notin \Delta$. Consider $S = \mathrm{pa}(i) \cup \mathrm{pa}^+(u) \setminus \{i\}$. Let $\pi$ be an open path given $S$. Since $\mathrm{pa}(i)$ are given, $\pi$ starts with $i \xrightarrow{\ell}$ edge. Let there exists a collider $k \in \mathrm{de}_\ell(i)$ on $\pi$, then some $z \in \mathrm{de}_\ell(k)$ must be in $S$ for $\pi$ being open. Then, $i \overset{\ell}{\rightsquigarrow} k \overset{\ell}{\rightsquigarrow} z$ implies that $z \notin \mathrm{pa}(i)$ due to acyclicity, and we have $z \in \mathrm{pa}^+(u)$. However, for tree-condition on $\mathcal{G}_\ell$, we must have $k = z = u$, and $\pi$ is blocked due to $i \xrightarrow{\ell} u \xleftarrow{\ell}$. Therefore, there is no such collider $k \in \mathrm{de}_\ell(i)$, and $\pi$ is a causal path $i \overset{\ell}{\rightsquigarrow} j$. Since $i \xrightarrow{1} u \overset{1}{\rightsquigarrow} j$ path is blocked by $u \in S$, $\pi$ is of the form $i \xrightarrow{2} v \overset{2}{\rightsquigarrow} j$. If $v \in \Delta$, $\pi$ is a causal path from $i$ to $j$ in which the child of $i$ is in $\Delta$, and the proof is completed. Suppose the contrary and let $v \notin \Delta$. Now, consider $S' = \mathrm{pa}(i) \cup \mathrm{pa}^+(u) \cup \mathrm{pa}^+(v) \setminus \{i\}$. Let $\pi'$ be an open path given $S'$, and without loss of generality, $\pi'$ starts with $i \xrightarrow{1}$. Note that $\pi'$ has to contain a collider $k \in \mathrm{de}_1(i)$ since the only causal path of the form $i \overset{1}{\rightsquigarrow} j$ is blocked by $u$. For $\pi'$ being open, some $z \in \mathrm{de}_1^+(k)$ is given in $S'$. We have $z \notin \mathrm{pa}(i)$ due to acyclicity. If we have $z \in \mathrm{pa}^+(u)$, tree-structured $\mathcal{G}_1$ implies that $k = z = u$, and $\pi'$ is blocked by $\mathrm{pa}(u)$. Finally, if $z \in \mathrm{pa}^+(v)$, $\mathcal{G}_1$ being a tree implies that $k = z = v$, and $\pi'$ is blocked by $\mathrm{pa}(v)$. Hence, $u \notin \Delta$ is not possible, and we have $u \in \Delta$, and the proof is completed.

## A.4 Proof of Lemma 1

We prove the desired result by investigating all possible cases. Since $i$ and $j$ are separable, at least one of them is not in $\Delta$. Without loss of generality, assume that $i \notin \Delta$.

(i) If $i \notin \mathrm{an}_1(j)$ and $i \notin \mathrm{an}_2(j)$: Consider $S = \mathrm{pa}(i)$. Suppose that $\pi$ is an open path between $\bar{i}$ and $\bar{j}$ given $\bar{S}$ on $\mathcal{G}_\mathrm{C}$. Since $\mathrm{pa}(i)$ is given, without loss of generality, $\pi$ starts with an edge $i \xrightarrow{1}$. Since $i \notin \mathrm{an}_1(j)$, $\pi$ contains a collider $k$ such that $i \overset{1}{\rightsquigarrow} k \xleftarrow{1}$. For $\pi$ to be open, a $z \in \mathrm{de}_1^+(k)$ must be in $S = \mathrm{pa}(i)$, which contradicts with acyclicity. Therefore, $S = \mathrm{pa}(i)$ is a separating set for $i$ and $j$.

(ii) If $i \in \mathrm{an}_1(j), i \notin \mathrm{an}_2(j)$, and $j \in \Delta$: Let $i \xrightarrow{1} u \overset{1}{\rightsquigarrow} j$. Since $i$ and $j$ are separable, by Theorem 3, $u \notin \Delta$. Consider $S = \mathrm{pa}(i) \cup \{\mathrm{pa}^+(u) \setminus i\}$. Suppose that $\pi$ is an open path between $\bar{i}$ and $\bar{j}$ given $\bar{S}$ on $\mathcal{G}_\mathrm{C}$. Since $\mathrm{pa}(i)$ is given, $\pi$ starts with an edge $i \xrightarrow{\ell}$. Since $u$ is given, $\pi$ contains a collider $k$ such that $i \overset{\ell}{\rightsquigarrow} k \xleftarrow{\ell}$. For $\pi$ to be open, a $z \in \mathrm{de}_\ell^+(k)$ must be in $S$. However, $z$ cannot be in $\mathrm{pa}(i)$ due to acyclicity. Also, $z$ cannot be in $\mathrm{pa}(u)$ since otherwise there would be multiple causal paths from $i$ to $u$ in $\mathcal{G}_\ell$, which contradicts the tree structure. Using the tree structure again, $\pi$ starts with $i \xrightarrow{\ell} u \xleftarrow{\ell}$. However, since $\mathrm{pa}^+(u)$ is given, $\pi$ cannot be open. Therefore, $S = \mathrm{pa}(i) \cup \{\mathrm{pa}^+(u) \setminus i\}$ is a separating set for $i$ and $j$.

(iii) If $i \in \mathrm{an}_1(j), i \notin \mathrm{an}_2(j)$, and $j \notin \Delta$: Let $i \xrightarrow{1} u \overset{1}{\rightsquigarrow} j$. We investigate this case in two subcases. First, suppose that $j \notin \mathrm{an}_2(i)$, and consider $S = \mathrm{pa}(j)$. Suppose that $\pi$ is an open path between $\bar{j}$ and $\bar{i}$ given $\bar{S}$ on $\mathcal{G}_\mathrm{C}$. Since $\mathrm{pa}(j)$ is given, $\pi$ starts with an edge $j \xrightarrow{\ell}$. Since $j$ is not an ancestor of $i$ in either graph, $\pi$ contains a collider $k$ such that $j \overset{\ell}{\rightsquigarrow} k \xleftarrow{\ell}$. For $\pi$ to be open, a $z \in \mathrm{de}_1^+(k)$ must be in $S = \mathrm{pa}(j)$, which contradicts with acyclicity. Therefore, $S = \mathrm{pa}(j)$ is a separating set for $i$ and $j$. Second, suppose that $j \in \mathrm{an}_2(i)$, and let $j \xrightarrow{2} v \overset{2}{\rightsquigarrow} i$. Since $i$ and $j$ are separable, by Theorem 3, at least one of $u$ and $v$ is not in $\Delta$. Without loss of generality, let $u \notin \Delta$ and consider $S = \mathrm{pa}(i) \cup \{\mathrm{pa}^+(u) \setminus i\}$. Following (ii), $S$ is a separating set for $i$ and $j$.

(iv) If $i \in \mathrm{an}_1(j), i \in \mathrm{an}_2(j)$, and $j \in \Delta$: Let $i \xrightarrow{1} u \overset{1}{\rightsquigarrow} j$ and $i \xrightarrow{2} v \overset{2}{\rightsquigarrow} j$. Since $i$ and $j$ are separable, by Theorem 3, $u \notin \Delta$ and $v \notin \Delta$. Consider $S = \mathrm{pa}(i) \cup \{\mathrm{pa}^+(u) \setminus i\} \cup \{\mathrm{pa}^+(v) \setminus i\}$. Since $\mathrm{pa}(i)$ is given, $\pi$ starts with an edge $i \xrightarrow{\ell}$. Since $u$ and $v$ are given, $\pi$ contains a collider $k$ such that $i \overset{\ell}{\rightsquigarrow} k \xleftarrow{\ell}$ without loss of generality. For $\pi$ to be open, a $z \in \mathrm{de}_\ell^+(k)$ must be in $S$. However, $z$ cannot be in $\mathrm{pa}(i)$ due to acyclicity. Also, $z$ cannot be in $\mathrm{pa}(u)$ (or $\mathrm{pa}(v)$) since otherwise, there would be multiple causal paths from

$i$ to $u$ (or $v$) in $\mathcal{G}_\ell$ which contradicts with the tree structure. Using the tree structure again, $\pi$ starts with $i \xrightarrow{\ell} u \xleftarrow{\ell}$ (or $i \xrightarrow{\ell} v \xleftarrow{\ell}$). However, since $\mathrm{pa}^+(u)$ (and $\mathrm{pa}^+(v)$) is given, $\pi$ cannot be open. Therefore, $S = \mathrm{pa}(i) \cup \{\mathrm{pa}^+(u) \setminus i\} \cup \{\mathrm{pa}^+(v) \setminus i\}$ is a separating set for $i$ and $j$.

(v) If $i \in \mathrm{an}_1(j)$, $i \in \mathrm{an}_2(j)$, and $j \notin \Delta$: Consider $S = \mathrm{pa}(j)$. Similar to the first subcase of (iii), $S$ is a separating set for $i$ and $j$.

We have exhausted all possible cases. In summary, if $i$ and $j$ are separable, then at least one of the sets $\mathrm{pa}(i)$, $\mathrm{pa}(j)$, $\mathrm{pa}(i) \cup \{\mathrm{pa}^+(u) \setminus i\}$, and $S = \mathrm{pa}(i) \cup \{\mathrm{pa}^+(u) \setminus i\} \cup \{\mathrm{pa}^+(v) \setminus i\}$ is a separating set for $i$ and $j$. Note that the size of these sets is upper bounded by $3d$ where $d$ denotes the maximum in-degree of a node in a component DAG.

### A.5 Proof of Lemma 2

We prove the four statements in order.

**Proof of Case 1.** There are three possible configurations for orientations of $(i - k)$ and $(j - k)$ pairs. We show that only $i \twoheadrightarrow k \twoheadleftarrow j$ is valid.

1. ($i \twoheadrightarrow k \twoheadleftarrow j$): By definition of the edge $\twoheadrightarrow$, without loss of generality, there exist paths $i \xrightarrow{1} u \overset{1}{\rightsquigarrow} k$ and $j \xrightarrow{2} v \overset{2}{\rightsquigarrow} k$ where $u, v \in \Delta$. Note that $u$ and $v$ are not necessarily distinct from $k$. Then, conditioning on $k$ opens the path

$$i \xrightarrow{1} u \xleftarrow{1} y \xrightarrow{2} v \xleftarrow{2} j \tag{15}$$

since $u$ and $v$ are colliders on the path and $y$ is a latent node. This implies that $k \notin S$, and this is a valid case.

2. ($k \twoheadrightarrow i$ and $k \twoheadrightarrow j$): Since $i \notin \Delta$ and $j \notin \Delta$, these orientations imply that $(i - k)$ and $(j - k)$ are not emergent edges. Then, we have $i \xleftarrow{\ell} k \xrightarrow{\ell} j$ for all $\ell \in \{1, 2\}$. Then, $k$ must be contained $S$ to separate $i$ and $j$, which contradicts with $k \notin S$, and this is not a valid case.

3. ($k \twoheadrightarrow i$ and $j \twoheadrightarrow k$): Without loss of generality and using Theorem 3, let $j \xrightarrow{1} u \overset{1}{\rightsquigarrow} k \xrightarrow{1} i$ in which $u \in \Delta$ and $k \xrightarrow{2} i$. Note that $u$ and $k$ are not necessarily distinct. Since $S$ separates $i$ and $j$, it contains a node $z \in \mathrm{de}_1^+(u)$. However, in this case, the path $j \xrightarrow{1} u \xleftarrow{1} y \xrightarrow{1} k \xrightarrow{1} i$ is opened since $k \notin S$. Therefore, this is not a valid case.

**Proof of Case 2.** There are two possible configurations for $(i - k)$. If $k \twoheadrightarrow i$, then $i \xleftarrow{\ell} k \xleftarrow{\ell} y \xrightarrow{\ell} j$ path cannot be blocked without conditioning on $k$, contradicting with $k \notin S$. Then, we have $i \twoheadrightarrow k$. Without loss of generality, let $i \xrightarrow{1} u \overset{1}{\rightsquigarrow} k$ for some $u \in \Delta$. Note that $S$ does not contain $k$. Otherwise $i \xrightarrow{1} u \xleftarrow{1} y \xrightarrow{1} j$ path would be open. Hence, this is a valid case.

**Proof of Case 3.** There are three possible configurations for orientations of $(i - k)$ and $(j - k)$ pairs. We show that two of them are valid.

1. ($i \twoheadrightarrow k$ and $j \twoheadrightarrow k$): Since $k \notin \Delta$, this implies $i \xrightarrow{\ell} k \xleftarrow{\ell} j$ for all $\ell \in \{1, 2\}$. Then, conditioning on $k$ opens the path $i \xrightarrow{\ell} k \xleftarrow{\ell} j$ for all $\ell \in \{1, 2\}$. This implies that $k \notin S$, and this is a valid case.

2. ($i \leftrightarrow\hspace{-0.3em}\twoheadrightarrow k$ and $j \leftrightarrow\hspace{-0.3em}\twoheadrightarrow k$): By using Case 2 of Theorem 3, without loss of generality, we have $i \xrightarrow{1} u \overset{1}{\rightsquigarrow} k$ and $k \xrightarrow{2} v \overset{2}{\rightsquigarrow} i$ for some $u, v \in \Delta$. There are two possibilities for $(j - k)$. First, $k \xrightarrow{1} z \overset{1}{\rightsquigarrow} j$ and $j \xrightarrow{2} w \overset{2}{\rightsquigarrow} k$ for some $z, w \in \Delta$. However, in this case, we have $i \xrightarrow{1} u \overset{1}{\rightsquigarrow} j$ and $j \xrightarrow{2} w \overset{2}{\rightsquigarrow} i$, which implies that $i$ and $j$ are inseparable by Theorem 3 and contradicts with the premise that $i$ and $j$ are separable. Then, we must have $k \xrightarrow{2} z \overset{2}{\rightsquigarrow} j$ and $j \xrightarrow{1} w \overset{1}{\rightsquigarrow} k$ for some $z, w \in \Delta$. In this case, conditioning on $k$ opens up the path $i \xrightarrow{1} u \xleftarrow{1} y \xrightarrow{2} w \xleftarrow{2} j$. This implies that $k \notin S$, and this is a valid case.

3. ($i \twoheadrightarrow k$ and $j \leftrightarrow\!\!\!\!\!\leftrightarrow k$): By using the Case 2 of Theorem 3, without loss of generality we have

$$j \xrightarrow{1} u \overset{1}{\rightsquigarrow} k \xleftarrow{1} i \quad \text{and} \quad i \xrightarrow{2} k \xrightarrow{2} v \overset{2}{\rightsquigarrow} j \tag{16}$$

where $u, v \in \Delta$. Since $k \notin S$, some $z \in \mathrm{de}_2^+(v)$ is given in $S$ to block the latter path. Then, for the path $i \xrightarrow{2} k \xrightarrow{2} v \xleftarrow{2} y \xrightarrow{1} u \xleftarrow{1} j$ to be blocked, $S$ does not contain any node in $\mathrm{de}_1^+(u)$. However, this implies that $j \xrightarrow{1} u \overset{1}{\rightsquigarrow} k \xleftarrow{1} i$ is open given $S$, which yields a contradiction. Hence, this is not a valid case.

**Proof of Case 4.** There are six possible configurations for orientations of $(i - k)$ and $(j - k)$ pairs. We show that three of them are valid.

1. ($i \twoheadrightarrow k$ and $j \twoheadrightarrow k$): Since $k \notin \Delta$, this implies $i \xrightarrow{\ell} k \xleftarrow{\ell} j$ for all $\ell \in \{1, 2\}$. Then, conditioning on $k$ opens the path $i \xrightarrow{\ell} k \xleftarrow{\ell} j$ for all $\ell \in \{1, 2\}$. This implies that $k \notin S$, and this is a valid case.

2. ($i \leftrightarrow\!\!\!\!\!\leftrightarrow k$ and $j \twoheadrightarrow k$): By Case 2 of Theorem 3, without loss of generality, we have $i \xrightarrow{1} u \overset{1}{\rightsquigarrow} k \xleftarrow{1} j$ and $j \xrightarrow{2} k \xrightarrow{2} v \overset{2}{\rightsquigarrow} i$ where $u, v \in \Delta$. Since $i$ is not an ancestor of $j$ in either graph, $i$ and $j$ can be separated. Also, conditioning on $k$ opens up the path $i \xrightarrow{1} u \xleftarrow{1} y \xrightarrow{1} j$. This implies that $k \notin S$, and this is a valid case.

3. ($k \twoheadrightarrow i$ and $j \twoheadrightarrow k$): Since $k \notin S$, without loss of generality $i \xleftarrow{1} k \xleftarrow{1} j$ path is open given $S$. Hence, this is not a valid case.

4. ($i \twoheadrightarrow k$ and $k \twoheadrightarrow j$): Since $k \notin \Delta$, we have $i \xrightarrow{1} k$, $i \xrightarrow{2} k$, and without loss of generality, $k \xrightarrow{1} u \overset{1}{\rightsquigarrow} j$ where $u \in \Delta$. To block the path $i \xrightarrow{1} k \xrightarrow{1} u \overset{1}{\rightsquigarrow} j$, some $z \in \mathrm{de}_1(u)$ must be conditioned on. However, doing so opens up to the path $i \xrightarrow{1} k \xleftarrow{1} u \xleftarrow{1} y \xrightarrow{1} j$. Hence, $i$ and $j$ cannot be separated without conditioning on $k$. Therefore, this is not a valid case.

5. ($k \twoheadrightarrow i$ and $k \twoheadrightarrow j$): The paths $i \xleftarrow{\ell} k \xrightarrow{\ell} u \overset{\ell}{\rightsquigarrow} j$, and $i \xleftarrow{\ell} k \xrightarrow{\ell} u \xleftarrow{\ell} y \xrightarrow{\ell} j$ cannot be blocked simultaneously without conditioning on $k$. Therefore, this is not a valid case.

6. ($i \leftrightarrow\!\!\!\!\!\leftrightarrow k$ and $k \twoheadrightarrow j$): Without loss of generality, we have $i \xrightarrow{1} u \overset{1}{\rightsquigarrow} k$ and $k \xrightarrow{2} v \overset{2}{\rightsquigarrow} i$ where $u, v \in \Delta$. For $k \twoheadrightarrow j$, there are two possibilities. If $k \xrightarrow{1} z \overset{1}{\rightsquigarrow} j$ for some $z \in \Delta$, we have $i \xrightarrow{1} u \overset{1}{\rightsquigarrow} k \overset{1}{\rightsquigarrow} j$. Subsequently, $i \twoheadrightarrow j$ which contradicts with $i$ and $j$ being separable. Therefore, we have $k \twoheadrightarrow j$ occur in the second graph as $k \xrightarrow{2} z \overset{2}{\rightsquigarrow} j$. Conditioning on $k$ opens up the path $i \xrightarrow{1} u \xleftarrow{1} y \xrightarrow{2} j$ since $k \in \mathrm{an}_1(u)$. This implies that $k \notin S$, and this is a valid case.

### A.6 Proof of Lemma 3

Recall that using mixture faithfulness assumption and Theorem 1, we have that $X_i \perp\!\!\!\perp X_j \mid X_S$ if and only if $i \perp\!\!\!\perp^{\mathrm{M}} j \mid S$ in composite DAG $\mathcal{G}_{\mathrm{C}}$. Then, if $(i - j) \in \mathbf{E}_{\mathrm{M}}$, either there is an edge between $i$ and $j$ in at least one component DAG or $(i - j)$ is an emergent edge. For the latter case, the inverse of the sufficient conditions for separability statements in Theorem 2 exactly gives the statements of this lemma.

### A.7 Proof of Lemma 4

If there is an edge between $i$ and $j$ in at least one of the component DAGs, then $X_i$ and $X_j$ are not independent for any conditioning set, and we have $(i - j) \in \mathbf{E}_{\mathrm{M}}$. The inverse of the necessary and sufficient conditions for separability statements in Theorem 3 exactly gives the statements of this lemma.

## B  Generalization to an Arbitrary Number of Component DAGs

In this section, we show that the results presented in Section 4 can be readily extended to a mixture of $m > 2$ DAGs. To this end, we extend all the definitions in Section 2 for $m$ component DAGs. We skip repeating the generalizations of simple definitions that are apparent from the context and just present the important ones here.

We consider $m$ component DAGs, $\{\mathcal{G}_\ell \triangleq (\mathbf{V}, \mathbf{E}_\ell) : \ell \in [m]\}$ with associated distributions $\{p_\ell : \ell \in [m]\}$. Then, the definition of $\Delta$ is generalized as

$$\Delta \triangleq \{i \in \mathbf{V} : \exists \ell, \ell' \in [m] \quad \text{such that} \quad p_\ell(X_i \mid X_{\mathrm{pa}_\ell(i)}) \neq p_{\ell'}(X_i \mid X_{\mathrm{pa}_{\ell'}(i)})\} . \tag{17}$$

Latent random variable $L \in [m]$ accounts for the true model underlying the observed data, where $L = \ell$ specifies that the true model is $p_\ell$. We denote the probability mass function (pmf) of $L$ by $q$. Then,

$$p_{\mathrm{M}}(x) \triangleq \sum_{\ell \in [m]} q(\ell) \cdot p_\ell(x) . \tag{18}$$

Accordingly, the definition of union edges is updated as

$$\mathbf{E}_{\mathrm{U}} \triangleq \{(i - j) : i, j \in \mathbf{V} , \ \exists \ell \in [m] : i \xrightarrow{\ell} j\} . \tag{19}$$

The definition of the edges of the mixture graph $\mathcal{G}_{\mathrm{M}}$ remains the same. The definition of the emergent edge becomes: *The edge $(i - j)$ in the mixture graph $\mathcal{G}_{\mathrm{M}}$ is called an emergent edge if the pair of nodes $i$ and $j$ are not adjacent in any $\mathcal{G}_\ell, \ell \in [m]$, but become inseparable in $p_{\mathrm{M}}$.* The set of emergent edges is given by

$$\mathbf{E}_{\mathrm{E}} \triangleq \{(i - j) : i, j \in \mathbf{V}, \ (i - j) \notin \mathbf{E}_{\mathrm{U}} \ \wedge \ \nexists A \subseteq \mathbf{V} \setminus \{i, j\} : \ X_i \perp\!\!\!\perp X_j \mid X_A\} . \tag{20}$$

We note that Theorem 1 is already given for a mixture of $m$ DAGs in (Saeed et al., 2020). Hence, we can still use it for inferring separation statements in composite DAG $\mathcal{G}_{\mathrm{C}}$.

For generalizing our separability result for general graphs (Theorem 2), we generalize its equivalent form Theorem 4 presented in Appendix A.

**Theorem 5** *Consider nodes $i, j \in \mathbf{V}$ such that $i$ and $j$ are not adjacent in any of the component DAGs, i.e., $(i-j) \notin \mathbf{E}_{\mathrm{U}}$.*

**Case 1)** *$i \in \Delta$ and $j \in \Delta$: $i$ and $j$ are always inseparable, i.e., $(i - j)$ is an emergent edge.*

**Case 2)** *$i \notin \Delta$ and $j \notin \Delta$: If $i$ and $j$ are inseparable, then there exist two component DAGs $\mathcal{G}_\ell$, $\mathcal{G}_{\ell'}$ such that $\mathcal{G}_\ell$ contains a $\Delta$-through path from $i$ to $j$ and $\mathcal{G}_{\ell'}$ contains a $\Delta$-through path from $j$ to $i$.*

**Case 3)** *$i \notin \Delta$ and $j \in \Delta$: If $i$ and $j$ are inseparable, then at least one component DAG contains a $\Delta$-through path from $i$ to $j$.*

**Proof** Note that Case 1 and Case 3 are identical to those of Theorem 4, and their proofs follow identically. Proof of Case 2 follows similarly as well. For $i \notin \Delta$ and $j \notin \Delta$, we have $\mathrm{pa}_\ell(i) = \mathrm{pa}_{\ell'}(i)$ and $\mathrm{pa}_\ell(j) = \mathrm{pa}_{\ell'}(j)$ for all $\ell, \ell' \in [m]$. Let the pair $(i - j)$ be inseparable and consider the set $S = \mathrm{pa}(i) \cup \mathrm{pa}(j)$. Then, there exists an open path $\pi$ between $\bar{i}$ and $\bar{j}$ given $\bar{S}$ on composite DAG $\mathcal{G}_{\mathrm{C}}$. Since $\mathrm{pa}(i)$ is given, any path that starts with $i \xleftarrow{\ell}$ is blocked for all $\ell \in [m]$. Without loss of generality, let $\pi$ start with an edge $i \xrightarrow{1}$. The case of $y \notin \pi$ follows identically to that of proof of Theorem 4. Let $y \in \pi$. Without loss of generality, suppose that $\pi$ contains $i \overset{1}{\rightsquigarrow} k \xleftarrow{1} \ldots y$ and $j \overset{2}{\rightsquigarrow} k' \xleftarrow{2} \ldots y$. For $\pi$ to be open, a $z \in \mathrm{de}_1^+(k)$ must be given in $S = \mathrm{pa}(i) \cup \mathrm{pa}(j)$, which implies that $k \in \mathrm{an}_1(j)$. Subsequently, $i \overset{1}{\rightsquigarrow} j$. Repeating the same line of arguments, we find that $j \overset{2}{\rightsquigarrow} i$. Finally, note that if none of the nodes on the path $i \overset{1}{\rightsquigarrow} j$ is not contained in $\Delta$, then we have the same path in $\mathcal{G}_2$ as $i \overset{2}{\rightsquigarrow} j$, which contradicts with $j \overset{2}{\rightsquigarrow} i$. Therefore, there exist $\Delta$-through paths between $i$ and $j$ in opposite directions in component DAGs $\mathcal{G}_1$ and $\mathcal{G}_2$.

Next, we generalize our separability result for a mixture of tree DAGs (Theorem 3).

**Theorem 6 (Separability in Tree Structures – Necessary and Sufficient Conditions)** *Suppose that $\mathcal{G}_1, \ldots, \mathcal{G}_m$ are tree-structured DAGs. Consider nodes $i, j \in \mathbf{V}$ such that $i$ and $j$ are not adjacent in any component DAG, i.e., $(i - j) \notin \mathbf{E}_{\mathrm{U}}$.*

**Case 1)** *$i \in \Delta$ and $j \in \Delta$: $i$ and $j$ are always inseparable.*

**Case 2)** *$i \notin \Delta$ and $j \notin \Delta$: $i$ and $j$ are separable if and only if there does not exist $\mathcal{G}_\ell$, $\mathcal{G}_{\ell'}$ such that the two DAGs contain $\Delta$-through paths between $i$ and $j$ in opposite directions such that the children of $i$ and $j$ on the paths are in $\Delta$.*

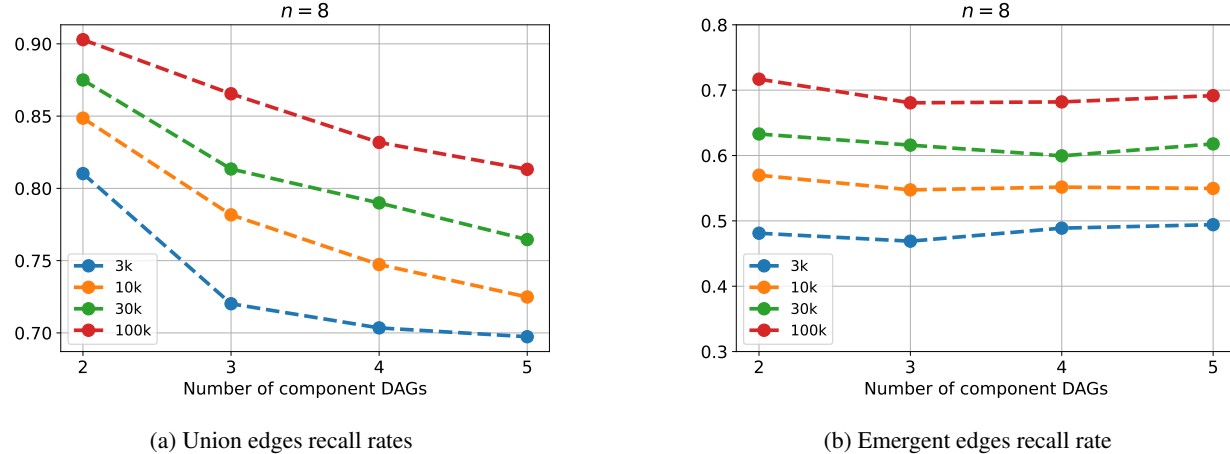

(a) Union edges recall rates

(b) Emergent edges recall rate

Figure 6: Skeleton recovery results for $n = 8$ at varying number of component DAGs and number of samples

**Case 3)** $i \notin \Delta$ and $j \in \Delta$: $i$ and $j$ are separable if and only if none of the component DAGs contains a $\Delta$-through path from $i$ to $j$ such that the child of $i$ on the path is in $\Delta$.

**Proof** The proof follows almost identically to the proof of Theorem 3 by setting $\ell = 1$ and $\ell' = 2$ without loss of generality, and using Theorem 5 instead of Theorem 4.

Finally, by using Theorem 6 instead of Theorem 3, Lemma 2 can be readily generalized to a mixture of $m$ tree-structured DAGs.

## C    Additional Simulations

### C.1    An Arbitrary Number of Component DAGs

In this section, we empirically support our claims in Section B that our approach can be generalized to a mixture of an arbitrary number of DAGs. To this end, we repeat the simulations in Section 5 for mixtures of $m \in \{2, 3, 4, 5\}$ component DAGs.

Figure 6a shows that the recall rate of union edges decreases as $m$ increases. This is not surprising since many union edges become *weaker* as $m$ increases in the sense that only $1/m$-th of all samples come from the model that the edge belongs to (unless the edge is shared over all components). That being said, the performance is still stronger than the recall rates of emergent edges, which we comment on next.

Figure 6b shows that there is no apparent change in the performance of recovering emergent edges in the skeleton of the mixture graph. This interesting observation can be explained by the effect of the number of component DAGs in two opposite directions. First, increasing $m$ also increases the size of $\Delta$ as more nodes are likely to be affected by the changes in the environment. However, the number of union edges also increases as $m$ increases. This, in turn, reduces the number of possible emergent edge pairs. As a result, the emergent recall rates remain around the same values as $m$ increases.

### C.2    The Choice of CI Test

In the experiments in Section 5, we have used a partial correlation test as a CI test, similar to the most closely related work to ours which also uses a partial correlation test even though the true mixture distribution is not Gaussian (Saeed et al., 2020). We note that there exist more sophisticated alternatives such as generalized covariance measure (GCM) (Shah & Peters, 2020), kernel-based conditional independence test (KCI) (Zhang et al., 2011), or Hilbert-Schmidt independence criterion (HSIC) (Gretton et al., 2007). However, the partial correlation test is much faster than these alternatives and usually achieves comparable performance (Mooij & Claassen, 2020). Still, out of comprehensiveness,

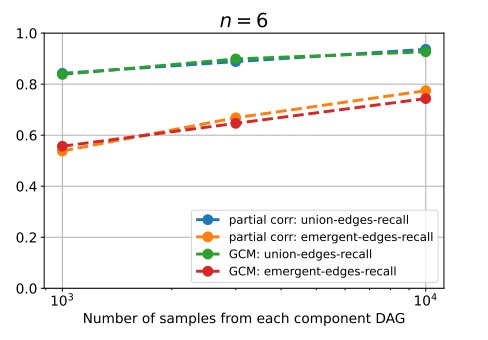 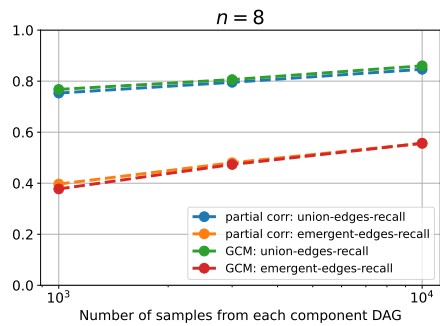

(a) Skeleton recovery results for $n = 6$         (b) Skeleton recovery results for $n = 8$

Figure 7: Comparison of partial correlation and GCM CI tests at skeleton recovery

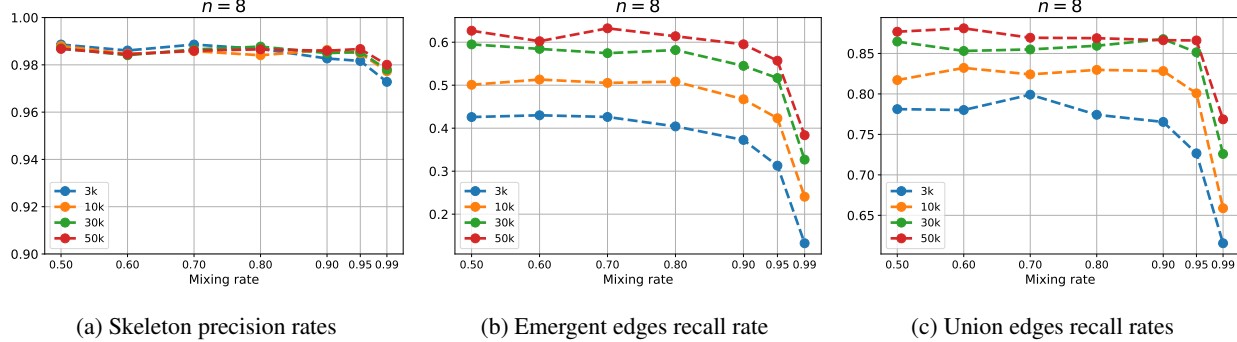

(a) Skeleton precision rates     (b) Emergent edges recall rate     (c) Union edges recall rates

Figure 8: Skeleton recovery results for $n = 8$ at varying mixing rate and number of samples

we repeat a subset of the simulations reported in Section 5 using GCM as CI test and compare the results with that of the partial correlation test.

We use the same setting as in Section 5 to generate mixture models and repeat the experimental procedure for 100 randomly generated tree pairs for $s \in \{1e3, 3e3, 1e4\}$ number of samples. Figures 7a and 7b show that there is no apparent difference in the performance of skeleton recovery while using either partial correlation or GCM tests. Therefore, the observations made in Section 5 remain the same when the partial correlation test is replaced with the GCM test.

## C.3 Varying the Mixing Rate

In Section 5, we have used an equal number of samples for each component DAG while creating the mixture model. In this section, we investigate the effect of varying this mixing rate. Specifically, we consider $n = 8$ nodes and total number of samples $s \in \{3e3, 1e4, 3e4, 5e4\}$ while varying the ratio of the dominant distribution in $\{0.5, 0.6, 0.7, 0.8, 0.9, 0.95, 0.99\}$. We repeat the experiments 500 times and report the average results for skeleton recovery in Figure 8.

Figure 8a shows that the very high precision rates are preserved even at extreme mixing rates of $0.95$ and $0.99$. The more important observation is regarding the recall rates. Figure 8b and Figure 8c show that the recall rates of both union edges and emergent edges are not significantly affected while varying the mixing rate from $0.5$ to $0.8$. However, when one of the models in the mixture becomes more dominant, indicated by a mixing rate over $0.9$, the performance decreases significantly. This is, in fact, not surprising due to the following reason: When one model becomes extremely dominant in the mixture, then two edge types become *weaker*, (i) the emergent edges that arise due to mixing of the components, and (ii) the edges exist in the minor model, which accounts for part of the union edges. Figure 8b illustrates the first observation.

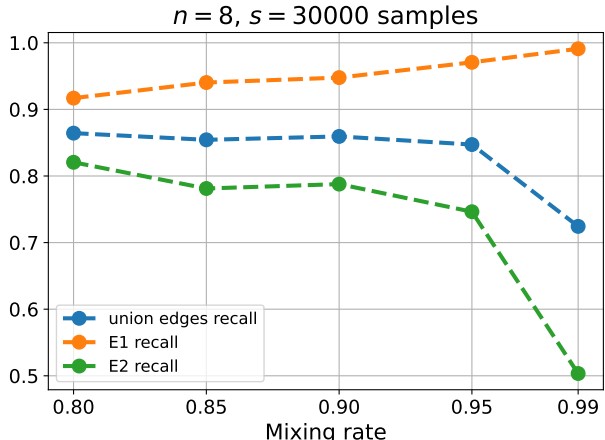

Figure 9: Skeleton recovery results for $n = 8$ at varying mixing rate and number of samples

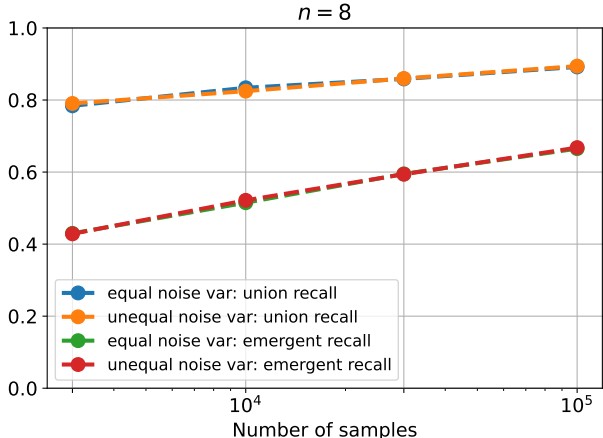

Figure 10: Skeleton recovery results for $n = 8$ at varying mixing rate and number of samples

Next, we scrutinize further to understand the behavior in Figure 8c better. We break down the union edges into $\mathbf{E}_1$ and $\mathbf{E}_2$ components, for which $\mathbf{E}_1$ corresponds to the edges that exist in the dominant model. Note that if an edge is shared in both models, then it is included in both $\mathbf{E}_1$ and $\mathbf{E}_2$. Figure 9 shows that, in accordance with our expectations, the performance degradation is due to the edges of the minor model. Specifically, recall rates of $\mathbf{E}_1$ become higher and recall rates of $\mathbf{E}_2$ become smaller as the ratio of the samples from $\mathcal{G}_1$ increases. This is not surprising since edges in $\mathbf{E}_2$ become weaker in the mixture model. This result can guide a practitioner on which approach to use when treating a mixture model, e.g., treating the model as a mixture if each component model is responsible for at least $10\%$ of the samples, or treating it is as a single model and dealing with weak edges using a different method if the minor model is responsible for only a few percent of the data samples.

## C.4 Changing the Noise Variance

In the simulations presented in the previous sections, we have used an equal noise variance $1$ as explained in the experimental setup. We repeat the same experimental procedure with unequal noise variances such that the noise variance of each node is sampled uniformly from $[0.5, 1.5]$. Figure 10 shows that the equal and unequal noise variance settings result in almost identical performances at the recovery of both emergent and union edges.

