# OpenReview forum: "Separability Analysis for Causal Discovery in Mixture of DAGs"
_TMLR — Accepted by TMLR_

### Review · Reviewer_mJ6F · 2023-11-02

**Summary Of Contributions:**

A good way to evaluate this paper is to start from the "Causal
Discovery Objective": "What causal relationships can be learned about
a mixture of DAGs .. ?" and work backwards. Given a conveniently
available CI oracle one can, of course, detect CI relations in the
mixture distribution (just as with any distribution) and represent the
presence/absence of these using an undirected graph (here called the
"mixture graph"). One could, of course, continue from here by ignoring
that this is a mixture distribution and continuing, say, with the
PC/FCI/RFCI algorithm. However, it is clearly more interesting to
attempt to uncover causal information in the component DAGs.

To this end the authors devise an orientation notation for edges in
the mixture graph and, more importantly, identify cases where these
orientations can be inferred from CI tests. Unsurprisingly, unless we
have additional knowledge - here about "\Delta" (the set of variables
with differing conditional distributions in the component DAGs) it
appears we cannot make further progress. With no knowledge of \Delta,
the authors do not provide a method for orientation - I suspect there
is none. In the case where we know some members of \Delta some
orientation is possible which represents a causal relation (Lemma 1).

The most important result is Theorem 1 about the composite DAG. Since
that is from an earlier paper the issue is how useful the additional
results are. The main contributions are Theorem 2 and Theorem 3. The
proofs of both these theorems work by analysing paths in the composite
DAG; the composite DAG and the assumption of "Mixture Faithfulness"
are thus key to the results in this paper. Theorem 2 gives sufficient
conditions for separability whereas Theorem 3 gives both necessary and
sufficient conditions when the component DAGs are tree DAGs and is
used to prove Lemma 1 which allows some orientation. The proofs of the
theoretical results in this paper are done by consideration of
different cases and then reasoning about open paths in the composite
DAGs. I found no errors (apart from a typo) and each individual part
of the proofs are not too complex. This case-by-case approach is
common in combinatorics, so not too surprising it is used here.

Building on the theoretical work an oracle-based constraint-based algorithm for learning "mixture graphs" is given.

**Audience:**

Yes

**Claims And Evidence:**

Yes

**Requested Changes:**

Although Lemma 1 is in Section 4.2 on Tree Mixtures I think the
restriction to tree DAGs should be explicitly given in its
statement. Since this is currently missing, strictly speaking Lemma 1
is incorrect.

Need to be clearer: "not adjacent (connected) in the individual DAGs":
does this mean not adjacent in all individual DAGs or in at least one
of them? Theorem 2 should be stated as "not adjacent in either component DAG"

The assumption of faithfulness should be stated earlier in the paper,
namely on p2. where inseparable pairs are defined to be "random variables
that cannot be made conditionally independent in the mixture
distribution." At present it is only mentioned parenthetically at the
bottom of p3 and properly stated at the bottom of p4.

Why not use some other test, e.g one that does not assume the true
distribution is Gaussian?

What is the true mixing rate (in the Experimental Setup)? Presumably
50/50? It would be interesting to see what happens as it varies.

In the proof of Theorem 2 "\neq P \Rightarrow \neg R" should be
"\neg P \Rightarrow \neg R"

**Strengths And Weaknesses:**

As well as allowing orientations I think the theoretical results also
have some independent interest to basically understand how "emergent
edges" between inseparable nodes arise. (One can also have emergent non-edges but these would be rare,
and are not considered in this paper.)

For Stage 1 of the algorithm (ie Algorithm1) we may have to consider
every possible conditioning set S to conclude that there should be an
edge in the mixture graph so this stage has exponential
complexity. Lemma 1 also requires that we know that k is not in "any
separating set of i and j". (These problems are noted in the
"Limitations" part of the Discussion.) So we don't have a really
practical algorithm. The paper would be improved if we could be given
some idea if there is a realistic prospect of developing the
"efficient algorithms" mentioned in Future Work. Also since we are
using an oracle here, at some point we will need to consider the
statistical problems that need to be addressed when learning from
(finite) data. The paper does not give a convincing case that this is an initial step to practical learning of DAG mixtures.

---

> ### Author Response · Authors · 2023-11-20
> **Author Response - Part 1**
>
> We thank the reviewer for the thoughtful evaluation and feedback. We have revised the manuscript to incorporate the suggested changes and address the weaknesses listed, a summary of which is provided below.
>
> ## Computational complexity
> In the revised manuscript, we have provided additional results on the computational complexity for tree mixtures and have provided additional comments about the general models, as summarized below.
> * **General graphs:** We note that for general graphs, Theorem 2 provides only sufficient conditions for the separability of two nodes. Without establishing matching worst-case necessary conditions, we cannot follow a PC-style approach to find the skeleton. Therefore, for general graphs, Stage 1 has $O(n^2 . 2^n)$ exponential complexity. We have added a discussion on this in Section 4.3.
> * **A Mixture of trees:** For a mixture of trees, Lemma 1 in the revised paper shows that if nodes $i$ and $j$ are separable, then there exists a separating set with size at most $3 d$ where $d$ denotes the maximum in-degree.  Therefore, we can modify Stage 1 to follow PC-style and achieve polynomial computational complexity $O(n^{3d + 2})$. Recall that PC has complexity $O(n^{d + 2})$.  The difficulty in our setting comes from having to block all paths between nodes $i$ and $j$ across multiple graphs, which may possibly contain causal relationships in opposite directions.
>
> * **Orientation phase:** In the revised paper, we have slightly updated the statement and proof of the orientation lemma for a mixture of trees. Specifically, the statement of Lemma 2 in the revised paper is as follows:
>
> “Suppose that $\mathcal{G}\_1$ and $\mathcal{G}\_2$ are tree-structured DAGs. Consider nodes $i \notin \Delta$ and $j \in \mathbf{V}$ that are separated by set $S$. Consider node $k\in \mathbf{V} \setminus \Delta$ for which $(i,k,j)$ is an unshielded triple in $\mathcal{G}\_{\rm M}$. Then, we have the following edge orientations in $\mathbf{E}\_{\rm M}$”
>
> * Note that unshielded triples and recorded separating sets for all separable node pairs can be directly read off from Stage 1 output. Therefore, Stage 2 does not perform any CI test (similar to the PC algorithm) and does not increase the computational complexity.
>
> * Finally, we agree with the reviewer that a comprehensive statistical analysis and algorithm development for practical causal discovery from finite data is important. We believe that the proposed adjustment and reaching polynomial complexity for a mixture of trees is a step toward the overarching objective of practical learning of a mixture of DAGs.
>
>
>
> ## The comments regarding clarity
> * *Lemma 1 statement*: We have adjusted the statement of Lemma 1 (now Lemma 2 in the revised paper) to explicitly state that the considered DAGs have tree structures.
> * On page 2, we have made it clearer: “.. node pairs that are not adjacent (connected) in all individual DAGs”. Similarly, we have adjusted the Theorem 2 statement as “.. not adjacent in either component DAG”.
> * *Faithfulness*: We added the following statement to the description of our framework on page 2: “We note that faithfulness -- that is, conditional independencies in the data are due to the separations in the true DAG -- is a core assumption in constraint-based causal discovery (Spirtes et al. (2000), Pearl (2009)). Hence, in this paper, we adopt a mixture faithfulness assumption, similar to the existing literature on causal discovery in a mixture of DAGs.”
> * We fixed the typo in the proof of Theorem 2: $\neg P \implies \neg R$.

---

> ### Author Response · Authors · 2023-11-20
> **Author Response - Part 2**
>
> ## Experiments
> * **CI test:** We have used a partial correlation test as a CI test for two reasons. (1) It is much faster than the alternatives (e.g., GCM, Kernel-based, HSIC). (2) It is widely used in the literature. For instance, the closely related work Saeed et al. also uses partial correlation. In a related context, Mooij et al. (JCI paper) reported that the partial correlation test is at least an order of magnitude faster compared to alternatives without an apparent loss of accuracy.
> * That being said, in Appendix C of the revised paper, we report results with an alternative test, the Generalised Covariance Measure (GCM, Shah and Peters, 2020). We observe no apparent change in the performance, as consistent with the existing literature, and the observations remain the same, e.g., emergent edges require more samples than union edges to achieve similar recall rates.
>
> Rajen D. Shah and Jonas Peters. The hardness of conditional independence testing and the generalized covariance measure. Ann. Statist. 48 (2020), no. 3, 1514--1538.
>
> * **Varying mixing rate:** For the experiments in Section 5, the mixing rate was 50/50. In Appendix C of the revised paper, we have further included results on the effect of the mixing rate. Specifically, we conducted simulations such that for the same number of total samples, the ratio of the dominant model ranges from 0.5 to 0.99. The key observations are summarized as follows.
>     * Recall rates of both union edges and emergent edges are not significantly affected while varying the mixing rate from 0.5 to 0.8. However, when one of the models in the mixture becomes too dominant, e.g., the mixing rate of 0.95 and above, the performance decreases significantly.
>     * This can be explained by the fact that two types of edges become weaker as the mixture distribution becomes more unbalanced: (i) emergent edges that arise due to the mixing of the components and (ii) the union edges that are exclusive to the minor model. We scrutinize the second case further and confirm that the recall rates of the dominant model’s edges become higher and recall rates of the minor model become smaller as the ratio of the samples from the dominant model increases. This result can guide a practitioner on which approach to use when treating a mixture model, e.g., treating the model as a mixture if the mixing effect is significant (e.g., more than 10% of the samples), or treating it as a single model and dealing with weak edges separately.
>
> * **Generalization to an arbitrary number of DAGs:** We note that we have shown that our results are not limited to a mixture of two DAGs and can be readily generalized to an arbitrary number of DAGs. Please refer to Appendix B for these results and Appendix C for additional experiments regarding this extension.

---

### Review · Reviewer_JUCG · 2023-11-05

**Summary Of Contributions:**

The paper considers learning causal DAGs from observational data in the case where the observed data comes from a mix of multiple underlying causal structures. Specifically, the paper is mostly an investigation into conditions where two edges are separable when they are nonadjacent in one or more of the underlying DAGs.

Sufficiency conditions are given for separability in general and necessary and sufficient conditions are given for tree structured DAGs. Finally, an algorithm for selecting and orienting edges (that can be oriented) is provided for both circumstances.

Some empirical results from simulations are provided.

**Audience:**

Yes

**Claims And Evidence:**

Yes

**Requested Changes:**

- In addition to mentioning the related work, can the authors clarify the novelty and relation of the results provided to prior work (specifically the theorems about separability with and without the tree-based assumption?
- Can the authors characterize the computationally complexity of their approach, whether it's occurring at the orientation phase only, whether a PC style approach can be used at the edge selection phase and what specifically about the approach leads to the computational difficulties?
- Can the authors list any contexts where it would be appropriate to make the tree-based assumption?

**Strengths And Weaknesses:**

Strengths
- The paper considers a topic that is not well explored
- The approach is novel to the best of my knowledge
- Theoretical conditions are provided for separability

Weaknesses
- The relation to prior work could be better explored. While prior work is mentioned, it's difficult to determine the specific differences and how related and novel this work is.
- While computational challenges are mentioned, they're not well described. It's not clear why in the first stage, a PC style approach could not be used to select edges (but the algorithm given is searching over all subsets). Furthermore, if this is the approach that is taken, it's not clear how the empirical results were generated for n=8.
- The empirical results could be made to convey more information - it's not clear what insights the reader should take away from the results. The ER assumption is also probably not realistic.
- It's clear that the tree-structured assumption makes the problem easier and the results more interesting, but it's not clear that one can realistically assume this in any contexts.

---

> ### Author Response · Authors · 2023-11-20
> **Author Response - Part 1**
>
> We thank the reviewer for the thoughtful evaluation and feedback. The suggested changes are incorporated into the revised manuscript, a summary of which and some clarifications are provided below.
>
> ## Comparison to the related work
> To lay the context for discussing the novelty, we note that there are three studies that are related to the scope of this paper: Saeed et al. (2020), Strobl (2022), and Spirtes (1995). The shared objective of these papers is to develop a graphical model to represent as many CI relationships in the mixture distribution as possible.
>
> In contrast to these, our objective is to characterize the conditions under which these CI relationships can emerge (or disappear). These conditions are uninvestigated and unknown. For instance, it is unknown under what conditions the node pairs that are separable in the constituent component graphs become inseparable in the mixture distribution — our emergent edges. Therefore, establishing the conditions for separability with respect to the properties of component DAGs (our Theorem 2 and 3) is the key novelty of the present work, and there is no counterpart in the literature. To highlight this, we provide a brief summary of the known results:
>
>
> * Spirtes (1995) shows that mixture distribution is Markov with respect to the fused graph (our definition 2). However, it does not provide separability theorems for a pair of nodes.
> * Strobl (2022) proposes a mixture graph of their own (also called “mother graph” in their preceding studies) such that the mixture distribution is Markov with respect to the proposed graph. However, this study does not provide counterparts of our separability results either. An algorithm is proposed which is exclusively designed for time-series data.
> * In the work most closely related to ours, Saeed et al. (2020) propose composite DAG (our definition 3), and their main result is given as Theorem 1 in our work. Specifically, they show that under an ordering assumption (coined as “poset compatibility”) assumption, the mixture distribution can be represented with a MAG (maximal ancestral graph), so that the classical FCI algorithm can be used to learn the graph.
> * However, they do not provide conditions for when an emergent edge arises either. Furthermore, their ordering assumption does not hold for a large class of a mixture of graphs. For instance, it rules out the formation of any cycles across two graphs.
> * In the present work, we drop this ordering assumption. For our most significant separability result (Theorem 3), we use the tree assumption. Hence, our approach can be interpreted as complementary to Saeed et al. (2020). We also note that none of the prior work specifically studies tree-structured DAGs.
>
> ## Computational complexity
> In the revised manuscript, we have provided additional results on the computational complexity for tree mixtures and have provided additional comments about the general models, as summarized below.
> * **General graphs:** We note that for general graphs, Theorem 2 provides only sufficient conditions for the separability of two nodes. Without establishing matching worst-case necessary conditions, we cannot follow a PC-style approach to find the skeleton. Therefore, for general graphs, Stage 1 has $O(n^2 . 2^n)$ exponential complexity. We have added a discussion on this in Section 4.3.
> * **A Mixture of trees:** For a mixture of trees, Lemma 1 in the revised paper shows that if nodes $i$ and $j$ are separable, then there exists a separating set with size at most $3 d$ where $d$ denotes the maximum in-degree.  Therefore, we can modify Stage 1 to follow PC-style and achieve polynomial computational complexity $O(n^{3d + 2})$. Recall that PC has complexity $O(n^{d + 2})$.  The difficulty in our setting comes from having to block all paths between nodes $i$ and $j$ across multiple graphs, which may possibly contain causal relationships in opposite directions.
>
> * **Orientation phase:** In the revised paper, we have slightly updated the statement and proof of the orientation lemma for a mixture of trees. Specifically, the statement of Lemma 2 in the revised paper is as follows:
>
> “Suppose that $\mathcal{G}\_1$ and $\mathcal{G}\_2$ are tree-structured DAGs. Consider nodes $i \notin \Delta$ and $j \in \mathbf{V}$ that are separated by set $S$. Consider node $k\in \mathbf{V} \setminus \Delta$ for which $(i,k,j)$ is an unshielded triple in $\mathcal{G}\_{\rm M}$. Then, we have the following edge orientations in $\mathbf{E}\_{\rm M}$”
>
> * Note that unshielded triples and recorded separating sets for all separable node pairs can be directly read off from Stage 1 output. Therefore, Stage 2 does not perform any CI test (similar to the PC algorithm) and does not increase the computational complexity.

---

> ### Author Response · Authors · 2023-11-20
> **Author Response - Part 2**
>
> ## Tree-structured DAGs
>  While not all-encompassing, causal trees are widely studied in causal discovery literature and have real-world applications, especially in biological networks. For instance, protein signaling pathways can be commonly modeled by causal trees. In particular, bi-partite causal graphs are used to model gene networks in which genes induce protein expressions, and the expressed proteins either inhibit or activate other genes (Kontou et al. (2016)). NF-κB protein signaling pathway, which activates mammalian immune system cells to produce antibodies against inflammation, is also modeled by trees (Lodish et al., 2004). Furthermore, causal trees are also shown to be computationally effective while still being useful to closely approximate more complex models (Acid and de Campos, 1994). We have included this discussion in Section 1.3 of the revised manuscript.
>
> Panagiota I. Kontou, Athanasia Pavlopoulou, Niki L. Dimou, Georgios A. Pavlopoulos, and Pantelis G. Bagos. "Network analysis of genes and their association with diseases." Gene 590, no. 1 (2016): 68-78.
>
> Ting Liu, Lingyun Zhang, Donghyun Joo, and Shao-Cong Sun. "NF-κB signaling in inflammation." Signal transduction and targeted therapy 2, no. 1 (2017): 1-9.
>
> Silvia Acid and Luis M. de Campos. “Approximations of causal networks by polytrees: an empirical study.” In Advances in Intelligent Computing - IPMU’94, 5th International Conference on Processing and Management of Uncertainty in Knowledge-Based Systems, volume 945 of Lecture Notes in Computer Science, pages 149–158. Springer, 1994.
>
>
> ## Experimental details
> * **Exponential complexity for $n=8$:** In our experiments for $n=8$ nodes, we have indeed searched over all subsets in edge search (Stage 1). To avoid prohibitive runtimes, we have used a partial correlation test as a CI test. We note that the partial correlation test is the common practice in the literature since it usually provides comparable results to more sophisticated tests (e.g., GCM, KCI, HSIC) while being significantly faster.
> * **Erdös-Renyi model:** We note that Erdös-Renyi (ER) model is the standard for simulations in causality literature. For instance, the most closely related work also uses ER (Saeed et al., 2020). To name a few other examples, the ER model is used in joint inference of multiple causal contexts (Mooij et al., 2020), hybrid learning approaches to causal discovery under interventions (Squires et al., 2020), and continuous optimization-based approaches to causal discovery (Zheng et al., 2018).
>
> Chandler Squires, Yuhao Wang, and Caroline Uhler, “Permutation-based causal structure learning with unknown intervention targets,” in Proc. Conference on Uncertainty in Artificial Intelligence, August 2020, pp. 1039–1048.
>
> Xun Zheng, Bryon Aragam, Pradeep K. Ravikumar, and Eric P. Xing. "Dags with no tears: Continuous optimization for structure learning." Proc. Advances in Neural Information Processing Systems, December 2018, Montreal, Canada.

---

### Review · Reviewer_73B5 · 2023-11-13

**Summary Of Contributions:**

This work is concerned with identifying the causal structure of data that is collected from a mixture of underlying distributions/causal structures. The authors present work complimentary to the approach in Saeed, et al. (2020). The key difference between the two is that the approach in Saeed requires an ordering of the constituent networks. The present work drops that assumption, replacing it with the assumption that each constituent graph is tree structured for some of the more substantive results. The authors describe identifiability conditions for edges, and describe when we should expect to see edges which are unidentified in the mixture graph despite being identified in the constituent component graphs. A constraint based learning algorithm is described for recovering the mixture graphs, and a small set of experiments run on some simpler synthetic data show promising results.

**Audience:**

Yes

**Broader Impact Concerns:**

I do not have broader impact concerns here.

**Claims And Evidence:**

Yes

**Requested Changes:**

(a) (minor) If the authors could provide some discussion for points (1)-(3) above that would be great
(b) (minor-moderate) It would be nice if there were some additional results provided on more challenging data simulation settings to glean more intuition above the propose behavior from empirical results.
(c) (very minor) There are some references that should probably be included:
    (i) "Causality with Gates" by John Winn addresses the problem here with a different approach (leveraging factor graphs).
    (ii) Both "Learning Mixtures of Bayesian Networks" by Thiessen, et al. and "Learning with Mixtures of Trees", by Meila & Jordan look at the problem of learning DAGs/ Tree structured graphs. They are both non-causal but are relevant enough they should probably receive a citation.

**Strengths And Weaknesses:**

Overall, I think this is an interesting work on a topic that is largely understudied in the community. The authors do a very nice job of clearly laying out the problem that is being considered, clearly describing the assumptions and delineating and contextualizing within the relevant recent work. I think the results provided here are things that can serve as a strong basis for future work to build on. The algorithm provided is sensible and relatively simple, and is also well described by the authors.

In terms of the weaknesses of the paper:
(1) (small) As acknowledged by the authors, the strongest results make an assumption of tree structured networks. This is a fairly common assumption, but it's not clear when we should expect this to hold especially since we need it to hold over _each constituent network_
(2) (small) There is an implicit assumption here that there is a "hard membership" model for the mixtures. This clearly can be sensible in many cases but not always (e.g., in the dynamical systems example it's fairly easy to imagine a smooth transitioning between states)
(3) (medium) The paper implicitly assumes, and explicitly tests against in the experiments, the case where there are just two components in the mixture. It would seem that unless there is a very large amount of overlap this solution (and perhaps any solution) would lead to a lot of emergent pairs.
(4) (medium) The experimental section is very limited. It would be useful if the authors could test against a larger number of components, unequal variances for the noise distributions, etc.

To summarize here, overall I think this a nice paper, and would like to see some small modifications to improve some of the takeaways for readers.

---

> ### Author Response · Authors · 2023-11-20
>
> We thank the reviewer for the thoughtful evaluation and feedback. We are also glad to hear that the reviewer finds our results form a strong basis for future work, which was among the key motivations of our work.
>
> We have revised the manuscript to incorporate the suggested changes and address the weaknesses listed, a summary of which is provided below.
>
> **Tree-structured DAGs:**  While not all-encompassing, causal trees are widely studied in causal discovery literature and have real-world applications, especially in biological networks. For instance, protein signaling pathways can be commonly modeled by causal trees. In particular, bi-partite causal graphs are used to model gene networks in which genes induce protein expressions, and the expressed proteins either inhibit or activate other genes (Kontou et al. (2016)). NF-κB protein signaling pathway, which activates mammalian immune system cells to produce antibodies against inflammation, is also modeled by trees (Lodish et al., 2004). Furthermore, causal trees are also shown to be computationally effective while still being useful to closely approximate more complex models (Acid and de Campos, 1994). We have included this discussion in Section 1.3 of the revised manuscript.
>
> Panagiota I. Kontou, Athanasia Pavlopoulou, Niki L. Dimou, Georgios A. Pavlopoulos, and Pantelis G. Bagos. "Network analysis of genes and their association with diseases." Gene 590, no. 1 (2016): 68-78.
>
> Ting Liu, Lingyun Zhang, Donghyun Joo, and Shao-Cong Sun. "NF-κB signaling in inflammation." Signal transduction and targeted therapy 2, no. 1 (2017): 1-9.
>
> Silvia Acid and Luis M. de Campos. “Approximations of causal networks by polytrees: an empirical study.” In Advances in Intelligent Computing - IPMU’94, 5th International Conference on Processing and Management of Uncertainty in Knowledge-Based Systems, volume 945 of Lecture Notes in Computer Science, pages 149–158. Springer, 1994.
>
> **Hard membership:** We thank the reviewer for pointing out the possibility of gradual, smooth transitions. The nature of analyzing such models will have a major difference from the setting we are considering. Specifically, tracking and identifying transitions from one model to another involves (1)  change-point detection and dynamic model tracking/estimating and (2) developing a counterpart of composite DAG for the new mixture distribution while still preserving Markov properties. We have added a summary of this discussion to Section 6 of the revised manuscript.
>
> **Generalization to a mixture of an arbitrary number of component DAGs:** The two-component mixture model was the main focus to facilitate clarity in exposition. However, our results can be readily generalized to a mixture of an arbitrary number of components. We have included a discussion on generalization in Appendix B in the revised paper.
>
>
> **Experiments:** Thank you for the suggestion for additional experiments. We have added additional results for the following experiments in Appendix C.
> * **A mixture of an arbitrary number of component DAGs:** We repeat the experiments for skeleton recovery for a mixture of $m \in \\{2,3,4,5\\}$ models and report the results in Appendix C. We summarize the observations below as follows.
>     * *Recall rates of the union edges:* The performance decreases as $m$ increases. This is expected since many union edges become weaker as $m$ increases in the sense that only $1/m$-th of all samples come from the model that the edge belongs to (unless the edge is shared over all components). That being said, the performance is still stronger than the recall rates of emergent edges, which we comment on next.
>     * *Recall rates of the emergent edges:* We observe no apparent change in the recall rates of the emergent edges as $m$ varies. This interesting observation can be explained by the effect of $m$ in two opposite directions. First, increasing $m$ also increases the size of $\Delta$ as more nodes are likely to be affected by the changes in the environment. However, the number of union edges also increases as $m$ increases. This, in turn, reduces the number of possible emergent edge pairs. As a result, the emergent recall rates remain around the same values as $m$ increases.
> * **Unequal noise variance:** We repeat the experiments with unequal noise variances such that the noise variance of each node is sampled uniformly from $[0.5,1.5]$. We observe that equal and unequal noise variance settings result in almost identical performances at the recovery of both emergent and union edges. Please refer to Appendix C for exact results.
> * Appendix C also contains new experiments that investigate (i) the effect of the mixing rate of the component models in the mixture and (ii) the choice of CI test.
>
> **Additional references:** We thank the reviewer for pointing out related work we have missed. We have added them to the related work in Section 1.3.

---

### Decision · Action_Editor_jXg9 · 2024-01-02

**Recommendation:** Accept as is

**Comment:**

The authors and reviewers engaged in discussion and the authors made changes to the manuscript to improve it based on the reviewer comments. The resulting manuscript is much improved and supported for acceptance by the reviewers.

**Audience:**

Some of the individuals in TMLR's audience would be interested in knowing the findings of this paper because it builds on recent work in learning causal graph structure.

**Claims And Evidence:**

This work claims a framework for recovering the causal graph structure for a mixture of DAGs. Building on the work of Saeed et at. 2020, the paper derives theorems on the conditions of separability in the DAGs. Finally, the paper claims an algorithm for learning the mixture graph. Empirical experiments on data generated from an Erdos-Renyi model support the claim of the algorithm efficacy which itself derives from the theoretical results. The reviewers found the claims well supported by the evidence.

---

> ### Author Response · Authors · 2024-01-10
> **Camera-ready revision**
>
> We thank the action editor and the reviewers for their thorough reviews and constructive feedback, which helped us improve the paper. We have uploaded the camera-ready revision.
>
> Sincerely,
>
> Authors